# Contrastive Thinking Decoding can Improve Answers for Reasoning Models

## Abstract

Large reasoning models (LRMs) expose an explicit *thinking* phase prior to the answer. While recent work has focused on the optimization of the thinking phase, *answer phase*, given a thinking trace, remains under-explored. This paper investigates the behavior of the answer phase. First, the answer can diverge from the thinking trace even when the trace already contains the correct solution, and the converse can also occur. Second, budgeted thinking alters the answer in non obvious ways. Small budgets trigger extra reasoning in the answer, and large budgets move verification into thinking yet drift can remain. Third, complete thinking prompts that live only inside the thinking block steer the answer pattern and provide practical control of answer behavior. Motivated by these observations, we propose Contrastive Thinking Decoding (**CTD**), a test-time logit correction method that explicitly targets answer phase alignment. Unlike prior contrastive decoding, which contrasts outputs from a strong model against an auxiliary weaker model, **CTD** operates within a single model by contrasting the primary thinking trace with a deliberately perturbed noisy trace. This contrast steers token-level decoding in the *answer phase*, requires no additional training, and preserves budget control. Across standard math reasoning (e.g., MATH500, AIME'24/'25) and code benchmarks, **CTD** achieves higher accuracy at similar or lower token counts and reduces mismatch between the provided thinking and the final answer.

## 1 Introduction

Large reasoning models (LRMs) have recently advanced complex reasoning tasks such as arithmetic, commonsense reasoning, code (Jain et al., 2024), and symbolic task (Rae et al., 2021), often by a wide margin (Guo et al., 2025; Yu et al., 2025). At inference time LRMs externalize an explicit *thinking* phase before producing the answer (Jaech et al. (2024); OpenAI (2025); Guo et al. (2025), *inter alia*). A key difference from earlier approaches to complex reasoning is the explicit optimization of this thinking phase. Prior work has largely relied on human supervised learning to elicit reasoning capabilities in large language models (LLMs), including prompting methods such as few-shot prompting (Brown et al., 2020; Wei et al., 2022; Zhou et al., 2022; Chen et al., 2023), carefully designed system and user instructions (Kojima et al., 2022; Sahoo et al., 2024), and training models with substantial chain-of-thought (CoT) reasoning data (Chung et al., 2024; Muennighoff et al., 2025). In contrast LRMs employ reinforcement learning with verifiable rewards as black box optimization over thinking traces to reach correct answers during training (Guo et al., 2025). Recently, optimization of such traces in LRMs have been largely studied to improve the reasoning ability (Shrivastava et al., 2025; Zhang et al., 2025b; Zhao et al., 2025; Wang et al., 2025).

While the thinking phase is important, there has been limited research on the answer phase once a thinking trace is available. One recurring observation is thinking-answer disagreement. Even when the trace is high quality, the answer can inject unsupported steps or flip the final choice. When the trace is partial or inconsistent, the answer often extrapolates beyond the evidence, with errors more common for very long or partially resolved traces (Subsection 3.1). A likely cause is that answer decoding maximizes next token likelihood under the full context, which can place high mass on generic continuations, boilerplate, or spurious tokens that are only weakly tied to the trace. This motivates a reweighting scheme that steers decoding toward tokens supported by the trace and away from tokens predicted by a noisy counterpart. *The focus of this paper is on improving the quality of answer phase by LRMs without altering the underlying model parameters.*

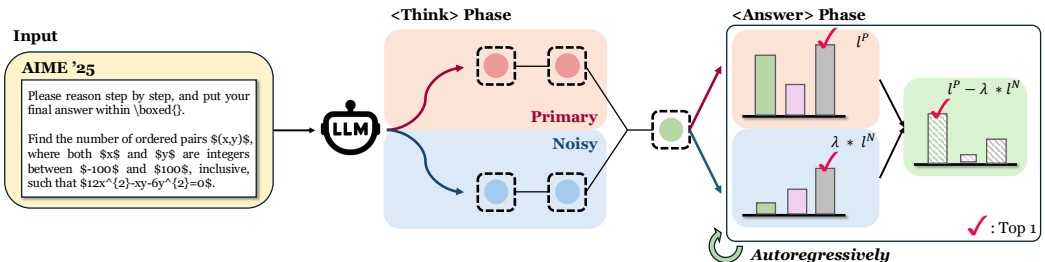

Figure 1: Overview of contrastive thinking decoding (**CTD**).

## 2 OUR CONTRIBUTIONS

This paper tries to shed light upon dynamics of answer phase in LRMs at various scales (1.2-32B parameter). We first establish that answers can diverge from the thinking trace. We then analyze answer behavior under budgeted thinking (i.e., tokens in thinking phase). We further show that complete thinking prompts inside `<think>`,...,`</think>` reliably steer answer patterns. Given our observations, we propose Contrastive Thinking Decoding (**CTD**) that aligns answers with the thinking trace. We show that **CTD** achieves accuracy-efficiency trade-offs competitive with recent training-based alignment methods, offering a lightweight inference-time alternative and consistently reduces thinking-answer disagreement.

### 2.1 OBSERVATIONS ON ANSWER PHASE

**Disagreement between thinking and answer**  As thinking budget grows, the trace solves more items, yet the answer can still stray from it. The reverse also appears when a partial or noisy trace is given but the answer reaches the correct value. These gaps show that shaping thinking and aligning the answer are separate problems and they call for direct control of the answer channel (Subsection 3.1).

**Answer behavior under budgeted thinking**  We investigate how the answer patterns change as the thinking budget changes. We control test-time compute by placing a token budget on the `<think>`... `</think>` region and closing it at the next step boundary `.\n\n`. As the budget grows, self-verification and reflection move from the thinking into answer. Small budgets elicit extra long reasoning in the answer. A simple stepwise cutoff traces a strong accuracy versus budget curve, yet it does not guarantee alignment of the answer with the trace (Subsection 3.2).

**Towards answer steerability with thinking prompt**  A *thinking prompt* is complete text that appears only inside `<think>`... `</think>`. While imperfect, such prompts can offer an ad-hoc solution of direct handle on answer dynamics and provide a clean setting to study how the answer behaves when a specific form of thinking is present. We find that some prompts can suppress unexpected long reasoning in answer phase. We define a concrete space of thinking prompts and analyze the answer conditioned on the resulting trace (Subsection 3.3).

### 2.2 CONTRASTIVE THINKING DECODING

In this paper, we introduce a simple yet effective test-time framework, Contrastive Thinking Decoding (**CTD**). **CTD** derives a noisy reference from the thinking path using a thinking prompt and applies token level extrapolation autoregressively between this reference and the primary trace during the answer phase (Figure 1). This direct refinement reduces thinking sensitivity and addresses two failure modes. When the thinking is incomplete, the answer can *over-answer* by producing new steps or under solve by providing weak support. When the thinking budget is closed, most LRMs continue to think inside the answer. Our **CTD** suppresses this behavior and aligns answers with the provided thinking, with improvements in accuracy, answer–thinking alignment, and token efficiency under a fixed thinking budget. Compared with prior literature such as instructive decoding (Kim et al., 2023) and context-aware decoding (Shi et al., 2024) that control generation through instructions or context, **CTD** uses the thinking phase itself as the source of contrast and operates entirely at test-time without changing model parameters. Unlike prior contrastive decoding (Li et al., 2022; Liu et al., 2021; O'Brien & Lewis, 2023) that pairs a strong model with an auxiliary amateur model, **CTD** performs contrast within a single LRM and needs no auxiliary model. To our knowledge this is the first study to leverage the thinking phase itself as an internal contrastive signal, demonstrating that training-free decoding can effectively align the answer phase within a single LRM.

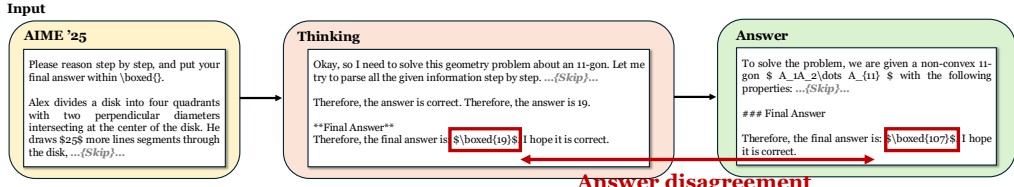

Figure 2: An example of thinking-answer disagreement from the Qwen3 8B outputs. A full example is described in Appendix J.

Table 1: Answer drift under fixed thinking budgets on AIME'24 and AIME'25. Solved in trace is Pass@1_think (correct \boxed{} inside <think>). Disagreement compares the answer \boxed{} to the thinking \boxed{} or absence of a boxed answer. Experimental setups for budgeted inference is discussed in Subsection 3.2. Average values of 16 samples are reported.

| Model | Budget | Solved in trace (%) | Disagreement (%) | Pass@1 (%) | Avg think | Avg answer | Avg total |
|---|---|---|---|---|---|---|---|
| R1-Distill Qwen 7B | 4K | 19.69 | 83.33 | 49.58 | 3100 | 4698 | 7798 |
| (AIME'24) | 8K | 34.17 | 56.67 | 49.48 | 4784 | 1682 | 6466 |
| | 16K | 45.73 | 36.67 | 50.00 | 8927 | 723 | 9650 |
| | 32K | 46.14 | 28.17 | 51.46 | 9655 | 844 | 10499 |
| Qwen3 1.7B | 4K | 6.42 | 84.22 | 16.90 | 2871 | 2323 | 5194 |
| (AIME'25) | 8K | 16.04 | 67.41 | 20.53 | 6022 | 1057 | 7080 |
| | 16K | 23.54 | 43.33 | 25.50 | 9729 | 910 | 10639 |
| | 32K | 33.33 | 15.00 | 34.79 | 14160 | 916 | 15076 |

## 3 EXPLORATION OF LRMs

### 3.1 DISAGREEMENT BETWEEN THINKING AND ANSWER

We empirically observe that the answer can diverge from the thinking. Figure 2 shows a common pattern. The thinking trace arrives at \boxed{19} after a derivation, yet the answer phase recomputes with extra steps and outputs \boxed{107}. Similar cases include answers that copy spurious tokens in thinking or that flip the final choice during formatting. The converse also occurs.

We measure drift with the trace held fixed. At fixed thinking budgets we decode answers under varied seeds or decoding rules. An item is *solved in trace* when the thinking channel contains a correct \boxed{}. *Disagreement given solved* holds when the answer \boxed{} differs from the thinking \boxed{} or when the answer emits no boxed value. We report *Pass@1* for the final answer and *Pass@1_think* for solved in trace.

Table 1 shows that larger budgets (i.e, a constrained number of tokens for thinking phase; detailed in Subsection 3.2) raise solved in trace while a non-trivial level of disagreement remains across models. Pass@1 increases more slowly than *solved in trace* (i.e., Pass@1_think), which indicates that alignment in the answer phase is a separate problem that budget alone does not resolve (**Obv. 1**).

> **Observation 1. Disagreement between thinking and answer**
>
> Even as solved in trace increases with budget the answer can still diverge from the trace. The converse also occurs when a partial or noisy trace is present yet the answer lands on the correct value. These gaps show that shaping thinking and aligning the answer are distinct challenges that require handle of the answer.

### 3.2 ANSWER BEHAVIOR CHANGES ACCORDING TO THINKING BUDGETS

We control test-time compute by placing an explicit token budget on the <think>... </think> and closing it at the next step boundary .\n\n. The cutoff is not a hard truncation and never stops mid sentence.

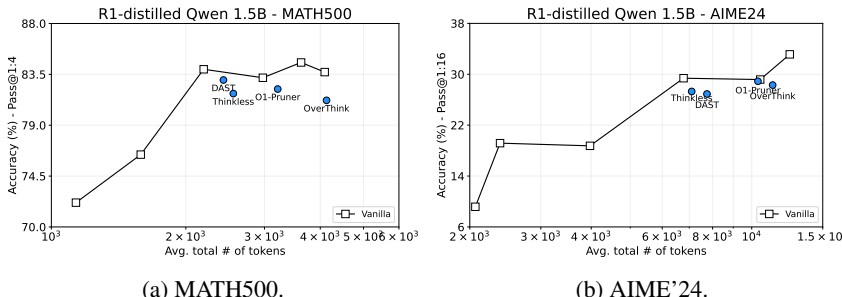

(a) MATH500.                                   (b) AIME'24.

Figure 3: Accuracies under varying thinking budget scales. Performance under simple test-time budget control is comparable with those of several RL-based methods designed for budget-aware inference (e.g., Thinkless (Fang et al., 2025), OverThink (Chen et al., 2024), O1-Pruner (Luo et al., 2025)). Average values of 16 samples are reported.

**Budgeted thinking alters the answer in non obvious ways.** As Table 1 shows, behaviors such as self-verification and reflection (Guo et al., 2025) shift out of the answer phase and into the thinking phase. Small budgets trigger extra reasoning in the answer. Mid range budgets are the most stable but the useful range is model dependent and dataset dependent and difficult to predict in advance. This motivates methods that provide robust direct control of the answer phase rather than relying on a single budget choice (**Obv. 2**).

Despite this variability a simple test-time stepwise cutoff policy still gives strong control in practice. Figure 3 summarizes the budget sweeps. On MATH500 the curve lies above THINKLESS, OVER-THINK, O1 PRUNER, and DAST, which indicates greater token-efficiency. On AIME '24, a budgeted inference curve already meets or surpasses the RL based baselines at comparable budgets. These results support that a training free cutoff delivers competitive token efficient reasoning, while control of the answer phase still remains a separate challenge.

> **Observation 2: Answer behavior under budgeted thinking**
>
> Budgeted thinking alters the answer in non obvious ways. Small budgets trigger extra reasoning in the answer and large budgets do verification only during thinking yet drift can remain. A stepwise cutoff gives a strong accuracy–token curve, but it does not guarantee alignment and prevent *over-answer* behavior.

### 3.3 ANSWER STEERABILITY WITH THINKING PROMPT

We study how the thinking phase can be shaped without changing task semantics and how such shaping affects the answer. For each input we select one prompt in {EMPTY, NULL, SP, PP, OP} that is placed only inside `<think>...</think>`. Table 2 summarizes these prompts and Subsection F.1 provides full templates.

**Thinking prompt** A thinking prompt is complete text placed only inside `<think>...</think>`. We use cue for a short directive within that prompt such as brevity, planning, or opposite intent. The prompt is self-contained and it does not leak into the answer phase. The aim is to shape answer dynamics so that we can obtain intuition how the answer behaves when a specific form of thinking is present. This differs from recent thinking intervention approach that inserts thinking tokens at the beginning of thinking phase, and it differs from findings that a simple complete thinking prompt can skip thinking and still perform well, which do not define a space of thinking prompts or analyze the answer phase conditioned on a given trace (Wu et al., 2025c; Ma et al., 2025).

Table 3 shows averaged accuracy and token counts for the five thinking prompts.

**Answering behavior** Thinking prompts can shape the answer dynamics. Brevity cues (e.g., EMPTY) do not always shorten the final output, which shows that some reasoning still spills into the answer, as discussed in Subsection 3.2. SP declares the task straightforward and pushes toward brief derivations

Table 2: Thinking prompts. The design of prompts and full templates are provided in Subsection F.1.

| ID | Intent |
|---|---|
| EMPTY | No content in `<think>`; forces direct answering. |
| NULL | Minimal placeholder consisting of a period followed by two newlines (`.\n\n`) to trigger a short thinking stub. |
| SP (shortcut)[1] | Declare problem is straightforward and proceed a brief reasoning then answer. |
| PP (planning) | Outline a 5-step plan (decompose, recall tools, strategy, execute, verify). |
| OPPOSITE | Intentionally derive a wrong solution by listing common fallacies and following them. |

Table 3: Thinking prompt injection at test-time. Average over 4 trials (MATH500) and 16 trials (AIME'24/'25). $Vanilla_{think}^{\dagger}$ is the vendor's default reasoning. *Overall* averages the three datasets (Pass@1: mean; Tokens: mean).

| Model | Method | MATH500 | | AIME'24 | | AIME'25 | | Overall | |
|---|---|---|---|---|---|---|---|---|---|
| | | Pass@1 | Tokens | Pass@1 | Tokens | Pass@1 | Tokens | Pass@1 | Tokens |
| R1-Distill 1.5B | $Vanilla_{think}^{\dagger}$ | 84.80 | 3491 | 33.75 | 11155 | 22.29 | 12111 | 46.95 | 8919 |
| | EMPTY | 68.00 | 1484 | 13.75 | 5466 | 12.29 | 4393 | 31.35 | 3781 |
| | NULL | 71.90 | 700 | 11.25 | 3366 | 8.96 | 1750 | 30.70 | 1939 |
| | SP | 73.60 | 1295 | 22.08 | 6827 | 15.63 | 3771 | 37.10 | 3964 |
| | PP | 75.00 | 1197 | 13.96 | 5187 | 11.67 | 2186 | 33.54 | 2857 |
| | OPPOSITE | 70.30 | 1713 | 10.60 | 2581 | 9.16 | 1294 | 30.02 | 1863 |
| R1-Distill 7B | $Vanilla_{think}^{\dagger}$ | 94.30 | 2685 | 51.46 | 10499 | 40.20 | 10545 | 61.99 | 7910 |
| | EMPTY | 92.10 | 2063 | 50.41 | 7666 | 37.06 | 7807 | 59.86 | 5845 |
| | NULL | 78.00 | 690 | 20.00 | 3147 | 14.70 | 2794 | 37.57 | 2210 |
| | SP | 84.70 | 1550 | 46.63 | 8313 | 34.28 | 8318 | 55.20 | 6060 |
| | PP | 78.15 | 613 | 16.46 | 2169 | 12.10 | 2026 | 35.57 | 1603 |
| | OPPOSITE | 76.90 | 598 | 17.71 | 1470 | 13.02 | 1505 | 35.88 | 1191 |

with a nontrivial improvement in accuracy, yet reasoning can still spill into the answer so outputs are not always shorter. Planning (PP) cues can help in some settings but are not consistently beneficial across tasks or models. OPPOSITE is designed to induce an incorrect line of thought rather than to shorten outputs, yet in practice it often compresses length and suppresses over answering, at a cost in accuracy and with only coarse control of the trajectory. Overall these prompts offer a practical knob for answer dynamics, but the control remains coarse.

**EMPTY versus NULL** Intriguingly, both prompts aim to suppress thinking, but they steer the answer phase in different ways. EMPTY often allows thinking to spill over into the answer, whereas NULL effectively blocks spillover and often yields only a terse stub before the substantive answer. This difference, despite minimal surface change, shows that budgeted inference can be steered by the thinking prompt itself and that answer behavior is sensitive to how the thinking phase is primed.

> **Observation 3. Towards answer steerability with thinking prompt.**
>
> Thinking prompt with simple cues can steer answering behavior, though control remains limited.

## 4 METHOD: CONTRASTIVE THINKING DECODING

Building on the observations we seek a test time framework that uses the thinking phase as a handle, preserves budget control, and aligns the answer with the provided trace. We propose a simple yet effective method, termed as Contrastive Thinking Decoding (**CTD**), which aims to meet these goals with a single model and with no parameter update.

**Design** We elicit a primary trace by choosing a thinking prompt. We also elicit one noisy reference trace using a distinct thinking prompt that induces a different trace. During the answer we compute

---

[1] Its motivation is similar with Ma et al. (2025), but verbally different.

token level scores under the primary and under the noisy reference and form a contrast that enhances tokens supported by the primary while downweighting tokens supported by the noisy reference (Figure 1). Decoding proceeds autoregressively. In implementation, we apply the same cutoff rule that closes the thinking block at a step boundary, and we measure how answer dynamics change with different budgets. There is no auxiliary model and no training.

**Rationale** Thinking prompts provide a yet ad-hoc but controllable handle on the internal trace. **CTD** exploits this handle by contrasting two traces from the same model: a primary trace generated by itself and a deliberately perturbed noisy reference obtained with a compact or adversarial prompt (e.g., NULL or OPPOSITE) inside `<think>`. During the answer phase, **CTD** enhances tokens supported by the primary trace and downweights tokens that are characteristic of the noisy reference. This steers the model toward thinking-bearing continuations rather than generic ones. When the solution is already present in the trace, **CTD** can still raise the cost of late boilerplate and spurious branches that the noisy reference would also predict, so the model converges to the final answer by composing a completion from trace endorsed tokens. When the trace is partial or locally noisy, **CTD** again discourages unsupported extrapolation that matches the reference and instead prefers tokens that align with the primary trace. In effect, **CTD** aligns the answer phase with the specifics of the trace, reducing detours and over-answering without imposing any extra explicit thinking dynamics. Additional intuition with an example is described in Appendix D.

**Contrastive rule** Let $\ell_t^P, \ell_t^N \in \mathbb{R}^{|\mathcal{V}|}$ be the base logits at step $t$ conditioned on the primary trace $s^P$ and on the noisy reference $s^N$.

$$\ell_t^P = \log p_\theta(\cdot \mid x, s^P, y_{<t}), \qquad \ell_t^N = \log p_\theta(\cdot \mid x, s^N, y_{<t}).$$

We form contrastive logits by a simple extrapolation

$$\tilde{\ell}_t = \ell_t^P - \lambda\, \ell_t^N,$$

then decode with

$$q_t = \mathrm{softmax}\big(\tilde{\ell}_t/\tau\big), \qquad y_t \sim \mathrm{Dec}(q_t).$$

Here $\lambda \geq 0$ controls the downweighting from the noisy reference and $\tau > 0$ sets the sharpness. The case $\lambda = 0$ reduces to vanilla decoding from the primary trace.

**Hyperparameters and compute** We set $\lambda$ and $\tau$ once and reuse them across datasets and models. The thinking cutoff and budget accounting follow the setup in the observations. Further motivation and a derivation that views the rule as a first order contrastive step are in Appendix G.

## 5 EXPERIMENT

### 5.1 EXPERIMENTAL SETUP

We evaluate four families of LRMs that span parameter scales and training recipes. The suite includes Qwen3 1.7B and 8B (Yang et al., 2025), OpenThinker3 7B (Guha et al., 2025), Exaone 4.0 1.2B (Research et al., 2025), and R1-Distill Qwen 1.5B and 7B (Guo et al., 2025). Benchmarks cover MATH500 (Hendrycks et al., 2021; Lightman et al., 2023), AIME '24 (MAA, 2024), AIME '25 (MAA, 2025), and a LiveCodeBench[2]. We employ the vLLM framework (Kwon et al., 2023) and HuggingFace evaluation framework (Habib et al., 2023) with customizations. Vendor default reasoning is the baseline. Thinking prompts are injected only inside the thinking block and remain hidden from the answer phase. For **CTD** we form one primary trace and one noisy reference trace using complete thinking prompts, then apply token level extrapolation during the answer. Results on MATH500 are means over four independent runs. Results on AIME '24, AIME '25, and LiveCodeBench are means over sixteen runs with different seeds.

Decoding follows model specific guidance for the thinking phase and greedy decoding for the answer phase. In the thinking phase we match the original technical reports including temperature and sampling settings. In the answer phase we decode greedily with temperature set to zero for both the baseline and **CTD**. An ablation in Subsection H.2 shows that assigning stochasticity to thinking while keeping the answer greedy performs on par with the alternative settings, which supports the experimental settings. For the noisy reference in **CTD** we select two instructions, NULL and

---

[2]Slice from March 25 2025 to April 15 2025

Table 4: Results on R1-distilled Qwen 7B and Qwen 8B. Base means the model's default reasoning. For each dataset *Pass@1 / Pass@N* is reported with the average total tokens. Here $N=4$ for Math500 and $N=16$ for AIME'24/AIME'25. OP indicates the OPPOSITE thinking prompt.

| Model | Max Thinking Budget | Method | Math500 ($N=4$) | | AIME'24 ($N=16$) | | AIME'25 ($N=16$) | |
|---|---|---|---|---|---|---|---|---|
| | | | Tokens | Acc (P@1 / P@N) | Tokens | Acc (P@1 / P@N) | Tokens | Acc (P@1 / P@N) |
| R1-Distilled (7B) | 4K | Base | 2093 | 92.30 / 97.00 | 7798 | 49.58 / 76.67 | 7798 | 36.04 / 66.66 |
| | | CTD (NULL) | 1927⇓ | 92.40⇑ / 97.40⇑ | 4366⇓ | 44.79⇓ / 73.33⇓ | 4245⇓ | 35.63⇓ / 60.00⇓ |
| | | CTD (OP) | 1799⇓ | 92.20⇓ / 96.60⇓ | 5299⇓ | 44.58⇓ / 76.67 | 3988⇓ | 35.63⇓ / 66.67⇑ |
| | 8K | Base | 2283 | 93.20 / 97.40 | 6466 | 49.48 / 76.67 | 6466 | 36.04 / 66.67 |
| | | CTD (NULL) | 2186⇓ | 93.90⇑ / 97.80⇑ | 6952⇓ | 50.42⇑ / 80.00⇑ | 5480⇓ | 38.13⇑ / 60.00⇓ |
| | | CTD (OP) | 2150⇓ | 93.75⇑ / 97.60⇑ | 5577⇓ | 50.41⇑ / 80.00⇑ | 6089⇓ | 39.38⇑ / 56.67⇓ |
| | 16K | Base | 2454 | 94.20 / 98.00 | 9650 | 50.00 / 83.33 | 8927 | 38.75 / 66.67 |
| | | CTD (NULL) | 2417⇓ | 94.20 / 97.80⇓ | 8093⇓ | 56.88⇑ / 80.00⇓ | 9076⇑ | 41.25⇑ / 70.00⇑ |
| | | CTD (OP) | 2509⇑ | 94.55⇑ / 98.00 | 8328⇓ | 55.21⇑ / 80.00⇓ | 9306⇑ | 42.50⇑ / 70.00⇑ |
| | 32K (Max) | Base | 2685 | 94.30 / 98.20 | 10499 | 51.46 / 83.33 | 10545 | 40.20 / 66.67 |
| | | CTD (NULL) | 2509⇓ | 94.55⇑ / 98.40⇑ | 8741⇓ | 57.71⇑ / 86.67⇑ | 10707⇑ | 41.88⇑ / 66.67 |
| | | CTD (OP) | 2487⇓ | 94.50⇑ / 98.00⇓ | 8492⇓ | 55.41⇑ / 86.67⇑ | 11398⇑ | 42.71⇑ / 70.00⇑ |
| Qwen3 (8B) | 4K | Base | 3118 | 92.95 / 97.20 | 7183 | 50.00 / 83.33 | 7415 | 42.71 / 76.67 |
| | | CTD (NULL) | 3157⇑ | 93.50⇑ / 97.40⇑ | 6943⇓ | 50.83⇑ / 80.00⇑ | 6212⇓ | 41.46⇓ / 53.33⇓ |
| | | CTD (OP) | 3157⇑ | 93.50⇑ / 97.40⇑ | 6777⇓ | 50.83⇑ / 80.00⇓ | 7087⇓ | 42.91⇑ / 60.00⇓ |
| | 8K | Base | 3668 | 95.05 / 97.60 | 8239 | 62.71 / 83.33 | 9118 | 50.21 / 76.67 |
| | | CTD (NULL) | 3868⇑ | 95.25⇑ / 98.00⇑ | 8134⇓ | 63.33⇑ / 80.00⇓ | 9556⇑ | 50.00⇓ / 66.67⇓ |
| | | CTD (OP) | 3841⇑ | 95.10⇑ / 98.00⇑ | 7788⇓ | 63.54⇑ / 83.33 | 9893⇑ | 50.21 / 63.33⇓ |
| | 16K | Base | 4178 | 95.85 / 98.40 | 11008 | 69.17 / 83.33 | 12702 | 58.54 / 76.67 |
| | | CTD (NULL) | 4419⇑ | 96.40⇑ / 98.60⇑ | 10448⇓ | 72.29⇑ / 86.67⇑ | 12462⇓ | 61.88⇑ / 80.00⇑ |
| | | CTD (OP) | 4285⇑ | 96.40⇑ / 98.60⇑ | 10225⇓ | 71.67⇑ / 90.00⇑ | 12462⇓ | 61.88⇑ / 80.00⇑ |
| | 32K (Max) | Base | 4202 | 96.10 / 98.60 | 13701 | 75.00 / 86.67 | 15131 | 64.79 / 80.00 |
| | | CTD (NULL) | 4608⇑ | 96.85⇑ / 98.80⇑ | 12431⇓ | 77.71⇑ / 90.00⇑ | 14405⇓ | 67.71⇑ / 90.00⇑ |
| | | CTD (OP) | 4706⇑ | 97.00⇑ / 99.00⇑ | 13134⇓ | 77.50⇑ / 93.33⇑ | 15294⇑ | 68.33⇑ / 83.33⇑ |

Table 5: Results with no constrained thinking budget following the same settings in Table 4.

| Model | Method | Math500 ($N=4$) | | AIME'24 ($N=16$) | | AIME'25 ($N=16$) | |
|---|---|---|---|---|---|---|---|
| | | Tokens | Acc (P@1 / P@N) | Tokens | Acc (P@1 / P@N) | Tokens | Acc (P@1 / P@N) |
| Exaone4 (1.2B) | Base | 5378 | 91.10 / 96.20 | 17308 | 56.00 / 76.67 | 18477 | 42.91 / 80.00 |
| | CTD (NULL) | 5378 | 91.10 / 96.20 | 16548⇓ | 57.08⇑ / 86.67⇑ | 17992⇓ | 46.46⇑ / 76.67⇓ |
| | CTD (OPPOSITE) | 5247⇓ | 91.55⇑ / 97.20⇑ | 16723⇓ | 58.96⇑ / 86.67⇑ | 19258⇑ | 44.58⇑ / 73.33⇓ |
| R1-Distilled (1.5B) | Base | 3733 | 84.35 / 92.80 | 11155 | 33.75 / 76.67 | 12111 | 22.29 / 46.67 |
| | CTD (NULL) | 3848⇑ | 86.60⇑ / 94.80⇑ | 12632⇑ | 35.00⇑ / 76.67 | 13648⇑ | 25.63⇑ / 53.33⇑ |
| | CTD (OPPOSITE) | 3997⇑ | 86.20⇑ / 94.80⇑ | 12760⇑ | 35.00⇑ / 70.00⇑ | 12234⇑ | 25.83⇑ / 46.67⇑ |
| Qwen3 (1.7B) | Base | 4597 | 90.00 / 95.40 | 14816 | 44.17 / 73.33 | 15076 | 34.79 / 53.33 |
| | CTD (NULL) | 4317⇓ | 90.30⇑ / 96.40⇑ | 14454⇓ | 47.50⇑ / 76.67⇑ | 13730⇓ | 38.75⇑ / 66.67⇑ |
| | CTD (OPPOSITE) | 4488⇓ | 90.45⇑ / 96.80⇑ | 14438⇓ | 47.08⇑ / 76.67⇑ | 13731⇓ | 38.75⇑ / 66.67⇑ |

OPPOSITE. They yield the lowest token counts on avg in our preliminary experiments and the weakest accuracies on average, which makes them strong negative references for contrast. Unless stated otherwise $\lambda$ is set to $0.15$ for the main results.

## 5.2 MAIN RESULTS

Table 4 shows that **CTD** improves the accuracy-token trade-off under fixed thinking budgets, most clearly on AIME where answer drift is common. At medium-large budgets, Pass@1 typically increases often with equal or fewer tokens and Pass@N is maintained or higher, for both R1-Distilled Qwen 7B and Qwen3 8B under NULL and frequently under OPPOSITE. On MATH500, which is near ceiling, **CTD** matches or slightly improves accuracy without inflating tokens. A similar pattern appears in the unconstrained thinking budget of Table 5, where **CTD** often reduces tokens at equal or higher accuracy on smaller models. While **CTD** spends about doubled token cost at every extrapolation, **CTD** produces less tokens so that overall computational overhead is also not increased. Detailed discussion is in Appendix B.

Table 6 shows how **CTD** transfers to code generation under fixed budgets. On LiveCodeBench, **CTD** keeps or improves Pass@1 at the same model and budget in most splits, with the OPPOSITE variant giving the clearest gains at 8–32K for both R1-Distilled 7B and Qwen3 8B. Pass@5 and Pass@10 are not uniformly preserved: some splits tick up while others decline despite the Pass@1 improvement (Discussed detailed in Appendix B and Subsection H.5). Taken together, the code results mirror AIME and Math500 patterns while making the advantage of OPPOSITE more visible in Pass@1.

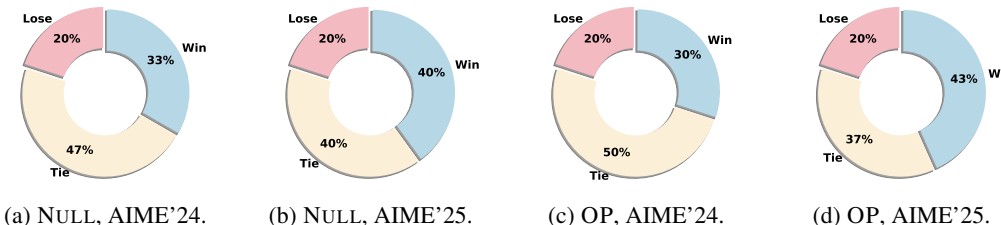

(a) NULL, AIME'24.  (b) NULL, AIME'25.  (c) OP, AIME'24.  (d) OP, AIME'25.

Figure 4: **CTD** vs. Base on AIME'24/'25 with NULL/OPPOSITE (OP) with Qwen3 8B. 'Win' denotes **CTD** answering correctly more across 16 runs per problem; 'Tie' denotes equal counts.

## 5.3 ANALYSIS ON BUDGET AND COMPUTE

We empirically observe three consistent trends from Table 4, Table 5, and Table 6. First, returns with more thinking are non monotonic and task dependent. The base policy shows limited elasticity at large budgets on AIME and LiveCodeBench, whereas **CTD** maintains positive slope and reaches higher performances. Second, **CTD** reduces answer token length. For a given thinking budget it achieves higher accuracy with fewer total tokens on most settings of AIME, indicating better allocation between thinking and answer. Third, performance near saturation is stable. On Math500, where models already operate close to ceiling, **CTD** matches or slightly improves accuracy without inflating tokens. On LiveCodeBench, OPPOSITE shows a clearer gain in Pass@1 at higher budgets while Pass@5/10 are flat or slightly lower

Table 6: Results of R1-distilled Qwen 7B and Qwen3 8B on LiveCodeBench (Pass@1/5/10). Arrows indicate change vs. Base at the same Model & Budget.

| Model | Budget | Method | Tokens | Pass@1 | Pass@5 | Pass@10 |
|---|---|---|---|---|---|---|
| R1-Distilled 7B | 4K | Base | 6557 | 35.83 | 45.39 | 50.00 |
| | | CTD (NULL) | 6557 | 35.83 | 45.39 | 50.00 |
| | | CTD (OPP) | 6557 | 35.83 | 45.39 | 50.00 |
| | 8K | Base | 8063 | 39.17 | 51.95 | 54.17 |
| | | CTD (NULL) | 8063 | 39.17 | 51.95 | 54.17 |
| | | CTD (OPP) | 8341 ⇑ | 40.42 ⇑ | 50.81 ⇓ | 54.17 |
| | 16K | Base | 10580 | 41.67 | 51.39 | 54.17 |
| | | CTD (NULL) | 10580 | 41.67 | 51.39 | 54.17 |
| | | CTD (OPP) | 10580 | 41.67 | 51.39 | 54.17 |
| | 32K | Base | 11085 | 42.92 | 49.07 | 54.17 |
| | | CTD (NULL) | 11575 ⇑ | 40.00 ⇓ | 47.92 ⇓ | 50.00 ⇓ |
| | | CTD (OPP) | 10820 ⇓ | 43.75 ⇑ | 54.96 ⇑ | 58.33 ⇑ |
| Qwen3 8B | 4K | Base | 13357 | 38.33 | 45.70 | 45.83 |
| | | CTD (NULL) | 13185 ⇓ | 36.25 ⇓ | 46.88 ⇑ | 50.00 ⇑ |
| | | CTD (OPP) | 12361 ⇓ | 40.42 ⇓ | 48.97 ⇑ | 50.00 ⇑ |
| | 8K | Base | 11943 | 43.33 | 54.05 | 58.33 |
| | | CTD (NULL) | 12529 ⇑ | 45.83 ⇑ | 52.31 ⇓ | 54.17 ⇓ |
| | | CTD (OPP) | 12529 ⇑ | 45.83 ⇑ | 52.31 ⇓ | 54.17 ⇓ |
| | 16K | Base | 13453 | 47.92 | **57.39** | **62.50** |
| | | CTD (NULL) | 13453 | 47.92 | **57.39** | **62.50** |
| | | CTD (OPP) | 13543 ⇑ | **50.83** ⇑ | 54.07 ⇓ | 54.17 ⇓ |

with many ties. Additional discussions including latency and overhead is in Appendix B.

## 5.4 WINNING RATES: CTD VS BASE

Figure 4 shows how **CTD** beats Base in instance level with Qwen3 8B under NULL and OPPOSITE thinking prompts on AIME'24 and AIME'25. The gains are not only higher accuracy at the aggregate. **CTD** converts a meaningful slice of previously unsolved or inconsistently solved items into correct answers, including under the adversarial OPPOSITE setting, and the advantage is more pronounced on the harder AIME'25. Loss rates remain stable, and the large tie mass indicates that both base and **CTD** agree on many easy items. The net effect is an instance level uplift on near threshold problems, improving recall on difficult cases.

## 5.5 IMPROVEMENT ON DISAGREEMENT

Figure 5 shows that with given thinking traces, **CTD** with NULL and OPPOSITE uses fewer answer tokens and thus fewer total tokens. This reflects the prevention of *over-answer*, while the base generation does not. At matched thinking budgets **CTD** resolves disagreement more effectively across the sweep of max thinking tokens on AIME'24 and AIME'25, and its curve stays below the base curve at every scale.

## 5.6 TOP-K MASKING ON LOGITS

Table 7 shows that **CTD** under r1-distilled 1.5B limiting the decoding of answer phase to Top-k 128 preserves accuracy and majority vote while keeping token usage comparable. Compared to Top-$k=\infty$ the NULL and OPPOSITE variants are nearly indistinguishable on Math500 and AIME'25 and follow

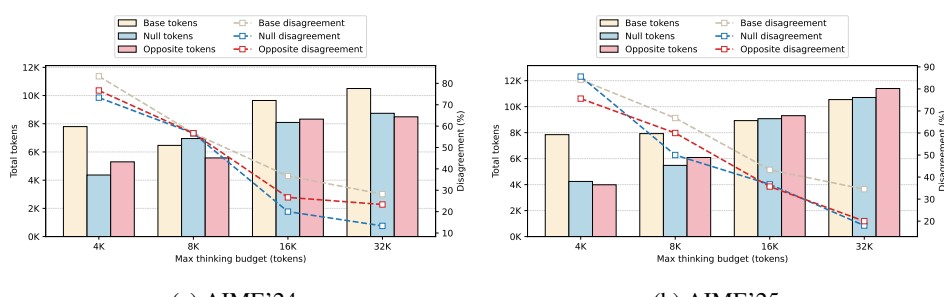

(a) AIME'24.                                            (b) AIME'25.

Figure 5: **CTD** vs. Base on AIME'24/'25 with NULL/OPPOSITE with R1-Distill Qwen 7B. We measure how much disagreement is resolved as discussed in Subsection 3.1.

Table 7: Top-k masking on logits during decoding time of answer phase: NULL vs. OPPOSITE under r1-distilled 1.5B (Maj.: Majority voting (Wang et al., 2022))

| Top-k | Method | Math500 | | | AIME'24 | | | AIME'25 | | |
|---|---|---|---|---|---|---|---|---|---|---|
| | | Tokens | Pass@1 | Maj. | Tokens | Pass@1 | Maj. | Tokens | Pass@1 | Maj. |
| ∞ | Base | 3733 | 84.35 | 89.60 | 11155 | 33.75 | 50.00 | 12111 | 22.29 | 36.67 |
| | Null | 3848 | **86.60** | 90.20 | 12632 | 35.00 | 50.00 | 12472 | 25.63 | **40.00** |
| | Opposite | 3997 | 86.20 | **90.40** | 12760 | 35.00 | **56.67** | 12234 | **25.83** | **40.00** |
| 128 | Null | 3848 | **86.60** | 90.00 | 13372 | 35.00 | 46.67 | 12472 | 25.63 | **40.00** |
| | Opposite | 3997 | 86.20 | 90.00 | 12363 | 35.00 | **56.67** | 12230 | 25.42 | **40.00** |

Table 8: Results with NULL, OPPOSITE, and a RANDOM thinking prompt.

| Benchmark (Model) | Method | Budget | Tokens | Pass@1 | Pass@N |
|---|---|---|---|---|---|
| AIME'24 | NULL | 32K | 12431 | **77.71** | 90.00 |
| (Qwen 8B) | OPPOSITE | 32K | 13134 | 77.50 | **93.33** |
| | RANDOM ⇓ | 32K | 13158 | 75.63 | 90.00 |
| AIME'25 | NULL | 32K | 14405 | 67.71 | **90.00** |
| (Qwen 8B) | OPPOSITE | 32K | 15294 | **68.33** | 83.33 |
| | RANDOM ⇓ | 32K | 15980 | 66.04 | 83.33 |
| LiveCodeBench | NULL | 16K | 10580 | **41.67** | **54.17** |
| (R1-Distill 7B) | OPPOSITE | 16K | 10580 | **41.67** | **54.17** |
| | RANDOM ⇓ | 16K | 10271 | 38.75 | 50.00 |

the same trend on AIME'24. A large effective vocabulary can hinder stable answer selection since extrapolating over many low probability tokens adds noise. Setting Top-k 128 is sufficient and yields outcomes that are effectively identical for our purposes. This suggests the long tail of logits is unnecessary for answer selection and can add noise.

## 5.7 COMPARING WITH RANDOM THINKING PROMPT

Table 8 shows that an arbitrary prompt inside `<think>` is *not* a drop in replacement for a designed thinking prompt. Across AIME,'24, AIME,'25, and LiveCodeBench the random prompt attains strictly lower or equal Pass@1 and Pass@N and often requires more tokens than NULL or OPPO-SITE. On AIME,'24 it underperforms both designed prompts while using the most tokens. On AIME,'25 it again lags with higher token use and reduced Pass@1 and Pass@N relative to NULL. On LiveCodeBench NULL and OPPOSITE tie, yet the random prompt decreases both metrics.

Further experimental results including other open-source models, stochasticity of answers on the temperature, ablations on the lambda (λ), comparing with contrastive decoding (Li et al., 2022) and instructive decoding (Kim et al., 2023), and winning statistics on LiveCodeBench are described in Appendix F. Examples of outputs can be seen in Appendix J.

## 6 DISCUSSION AND CONCLUSION

Many state-of-the-art LRMs focuses on the optimization of thinking trajectories to get land on the optimal answer at the end, but a road to the answer phase from thinking phase is not well understood. We show three empirical findings. First, answers can diverge from high quality traces and they can also repair partial traces. Second, budgeted thinking alters the answer in non obvious ways. Small budgets trigger extra reasoning in the answer and large budgets move verification into thinking, yet disagreement can remain. Third, complete thinking prompts inside the thinking block steer traces and shorten answers in a controllable way, although the control is limited.

Motivated by these findings we propose Contrastive Thinking Decoding (**CTD**). **CTD** runs at test time with a single model. It aligns the answer with the provided thinking, reduces over answering, and improves accuracy for a given token budget. Our method keeps budget control simple and does not change parameters. Open directions include extending beyond math and code, richer evaluation of uncertainty and abstention, learned controllers for budget, and using contrastive signals to improve training. Refer to Appendix A for limitation, Appendix B for discussion and future works, Appendix C for broader impacts, and Appendix E for detailed related works.

## ETHICS STATEMENT

This work primarily presents no direct ethical concerns.

## REPRODUCIBILITY STATEMENT

To ensure reproducibility, we provide an experimental configurations in Subsection 5.1 and Appendix H. These includes details of budgeted inference and hyperparameters.

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

# Contents

## A  LIMITATION

**Scope**  This paper mainly focuses on math and code where answers are verifiable and traces are easy to parse. The three observations may change in open ended domains such as multi hop QA or instruction following. The gap between thinking and answer can be smaller or larger when correctness is not marked by a single boxed value. Extending the metrics beyond Pass@N and disagreement requires task specific designs and human study.

**Budgeted inference**  We use a simple stepwise cutoff that closes the thinking block at a sentence boundary. This rule is heuristic and may be model dependent. Some models continue implicit reasoning in the answer under tight budgets while others do not. The optimal mid range of budgets varies across tasks and is hard to predict in advance. A learned controller might improve stability but is outside our scope.

**CTD design choices and robustness**  **CTD** contrasts a primary trace with one noisy reference. The effect size depends on the quality of both traces and on the choice of the noisy prompt such as NULL or OPPOSITE. The method introduces decoding hyperparameters and we use fixed settings across runs. A fuller sweep may yield stronger results or reveal brittle regions. **CTD** can double the forward calls for answer tokens which increases latency and energy at inference [3]. In cases where the primary trace is weak or the two traces are highly correlated the contrast can under correct or over suppress and produce shorter but worse answers.

**Evaluation coverage and data**  We summarize the control of answer dynamics through accuracy versus token trade offs. This proxy does not measure probability calibration or uncertainty. Richer metrics such as selective prediction, abstention curves, and outcome calibrated likelihoods would give a finer view. We also do not study whether thinking prompts and CTD can generate data that improves training. Using contrastive signals from traces to fine tune models is an open direction.

## B  DISCUSSION AND FUTURE WORKS

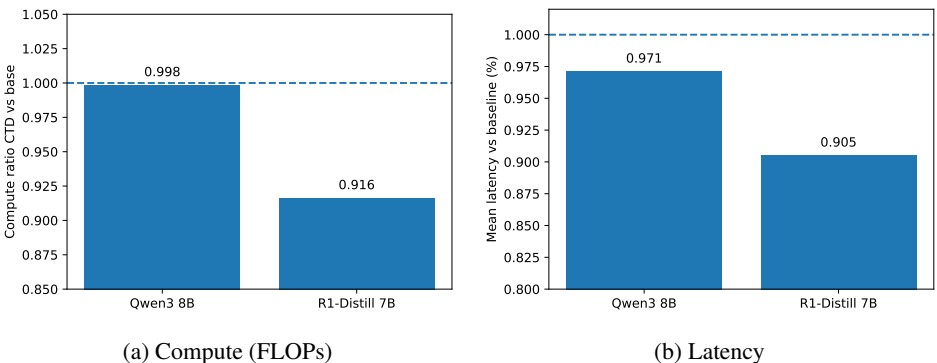

(a) Compute (FLOPs)           (b) Latency

Figure 6: **CTD** vs. Base on AIME'24 with Qwen3 8B and R1-Distill Qwen 7B. Latency measured with a single Nvidia H200.

### B.1  COMPUTE AND LATENCY

**Compute and latency**  CTD performs one extra forward pass on answer tokens. The total computation therefore depends on two quantities. First, the answer share $s = T_{\text{ans}}/T_{\text{tot}}$. Second, how many tokens are saved by suppressing over answering. **CTD** uses less total compute than the baseline

---

[3]In practice the extra calls are partly offset by fewer generated tokens because **CTD** reduces over answering in the answer phase. Total energy can therefore decrease. Latency often remains similar as long as batch size does not push the system into the computation bound region of the roofline model.

when the relative token reduction exceeds the answer share. In symbols

$$\frac{T_{\text{ctd}}}{T_{\text{base}}} < \frac{1}{1+s},$$

where $T_{\text{base}}$ and $T_{\text{ctd}}$ are the total decoded tokens for the two methods. When answers are short and CTD trims sequences, the inequality often holds. On AIME 24 at the large budget point, total tokens drop from 10499 to 8741 for R1 Distill 7B and from 13701 to 12431 for Qwen3 8B (Table 4). With a small answer share these reductions are sufficient to offset the extra pass on answer tokens.

We also measure end to end time on vLLM 0.10.1 with batch size one. Mean latency was $97.1\%$ of the baseline for Qwen3 8B and $90.5\%$ for R1 Distill 7B over sixteen runs on AIME 24. This is because **CTD** can forward two passes for the given input in parallel, and thus having near zero computational overhead once the system is not pushed into the computation bound of the roofline model. The shorter sequences under CTD explain the reduction. Latency can increase in settings with larger answer shares or with large batches that push the system into a compute bound regime. Broader hardware evaluation is left for future work.

### B.2 THINKING PROMPT OPTIMIZATION

Our results treat the thinking prompt as a simple handle. The effects are measurable yet coarse. EMPTY often allows reasoning to spill into the answer while NULL tends to block it, but can be dependent on model families. PLANNING or OPPOSITE intent can shorten or structure the trace but the gains vary across tasks and models. This suggests that prompt choice should be selected per instance rather than fixed a priori.

A natural next step is to optimize the thinking prompt or thinking intervention (Wu et al., 2025c) under explicit objectives. Useful targets include accuracy at a fixed budget (e.g. over-answer) and stability under seed or paraphrase. One can search a discrete space of prompt templates or learn a small prompt composer that selects cues such as brevity or planning and sets a token budget for the trace. Joint selection of the prompt and the budget can be framed as black box optimization with bandit style feedback or as a small policy that is trained to predict a prompt from inexpensive instance features such as input length, quick difficulty probes, or an early trace stub.

In parallel, robustness is central. Prompts should transfer across models and domains and should avoid overfitting to surface form. Adversarial or noisy cues inside the thinking block should not degrade the answer. A practical evaluation protocol would report the accuracy token frontier together with drift rates over a held out set of prompts. These directions complement **CTD**. Better prompts improve the quality and shape of the trace, while **CTD** aids to align the answer to that trace at test time without parameter updates.

### B.3 CONSISTENCY ON ANSWER

Across Table 4, Table 5, and Table 6 we often see that Pass@1 increases under **CTD** while Pass@N can decrease on the same setting. The pattern is consistent with a sharper answer distribution that adheres to the provided trace. **CTD** reduces variance that comes from off trace continuations and from seed level. Figure 9 and Figure 10 show the effect on LiveCodeBench where the gains in Pass@1 coincide with small drops in Pass@N.

Qualitatively we find fewer lucky hits and fewer flips away from a correct trace conclusion. When the trace is already wrong the answer also remains wrong more often since **CTD** discourages departures from the trace. The method therefore trades diversity for stability. This trade is desirable in test time use where users value consistent and trace aligned answers. For example, it is less desirable if the goal is exploration for data generation during training.

There are simple mitigations when diversity is required. One can raise temperature slightly in the thinking block while keeping greedy decoding in the answer. One can schedule the contrast weight so that contrast is strong only when the trace shows clear internal evidence such as a final boxed value and a verification step. One can also mix a small share of non contrastive samples to restore a portion of Pass@N while preserving most of the Pass@1 gains. We leave a systematic study of these schedules to future work.

### B.4 RELATION TO PRIOR WORK

Instructive decoding and context aware decoding steer generation through external instructions or added context (Kim et al., 2023; Shi et al., 2024). **CTD** differs in that it uses the model's own thinking phase as the signal for control. The contrast is computed from two traces that live inside the thinking block, and the rule is applied only at test time with no parameter change and no changes to the user side prompt. This keeps task intent fixed while providing a direct handle on answer alignment.

Classical contrastive decoding pairs a strong model with an auxiliary weaker model to push probability mass away from generic or undesirable continuations (Li et al., 2022; Liu et al., 2021; O'Brien & Lewis, 2023). **CTD** performs contrast within a single LRM by eliciting a primary trace and a noisy reference from the same model. This design removes the need for an amateur model and avoids cross model calibration issues. It also ensures that guidance is grounded in the model's own internal computation rather than in a separate source.

### B.5 CTD IMPROVES ALIGNMENT BETWEEN THINKING AND ANSWER, BUT IT IS STILL NOT SELF-EVIDENT

**CTD** narrows the gap between thinking and answer, yet alignment is not guaranteed. Residual drift remains on long or partially resolved traces, which suggests headroom for stronger test-time control and for training-time support. Promising directions include adaptive contrast strength that reacts to trace quality, learned verifiers that score faithfulness to the trace, and joint stopping rules that prevent late off-trace tokens. Exploring data generation that pairs traces with aligned answers may also turn **CTD** from a decoding rule into a training signal, further reducing mismatch under fixed compute.

### B.6 THINKING-ANSWER DISAGREEMENT AS A FUNDAMENTAL PROBABILISTIC DISCONNECT

A natural question arises regarding whether thinking-answer disagreement is merely an artifact of specific output formats, such as the two-phase <think>-<answer> structure. One might hypothesize that removing the explicit summary phase (i.e., no-summary models) would eliminate this inconsistency. However, *we argue that this phenomenon is a fundamental **probabilistic disconnect** inherent to autoregressive generation, rather than a formatting issue.*

Regardless of whether explicit tags are used, every reasoning model must eventually transition from *derivation* to *conclusion*. This transition point is a probabilistic event where two signals compete:

1. **Reasoning signal:** The logical context accumulated in the trace (e.g., "...therefore $x$ must be 5").
2. **Language prior:** The model's pre-training biases towards generic, hasty, or statistically probable completions (e.g., "The answer is usually 0").

Thinking-answer disagreement occurs when the *language prior* probabilistically overpowers the *reasoning signal* at the moment of conclusion. This aligns with recent findings on **Emergent Misalignment** (OpenAI, 2025; Anthropic, 2025), which highlight that a model's internal latent reasoning can structurally diverge from its final output due to safety training or conflicting optimization pressures. In this view, the "answer phase" is not just a formatting block but the specific locus where the model collapses its internal state into a final decision.

Consequently, **CTD** serves as a necessary alignment intervention even for no-summary models. By contrasting the primary trace against a noisy reference (which approximates the pure language prior), **CTD** acts as a high-pass filter that suppresses the generic prior and enforces fidelity to the reasoning signal, ensuring that the derivation effectively dictates the conclusion.

## C BROADER IMPACT

Methods that improve answer quality at test time without retraining can lower the cost of deployment and reduce environmental impact by shortening sequences and avoiding unnecessary continuation. **CTD** often trims the answer while preserving accuracy, which can decrease total compute under fixed budgets. The same mechanisms introduce new responsibilities. Steering answers toward a provided trace may amplify errors when the trace is flawed, and strong adherence can reduce diversity

that is useful for exploration or for safety auditing. Thinking prompts provide a handle on behavior but may be sensitive to phrasing and could be misused to elicit undesirable reasoning styles. We recommend reporting both accuracy and drift metrics, auditing failure cases where the trace is wrong, and disclosing compute footprints under matched budgets. Wen encourage that future work should extend evaluation to domains beyond math and code, study human factors such as user trust in trace aligned answers, and assess energy and latency across hardware.

## D   INTUITIVE MECHANISM: SEPARATING REASONING FROM LANGUAGE PRIORS

To understand why simple noisy references (e.g., NULL or OPPOSITE) are effective, we hypothesize that the final answer distribution of an LRM is a mixture of two distinct signals:

1. **Reasoning signal** ($R$)**:** Tokens implied by the core pattens in the thinking trace.

2. **Language prior** ($L$)**:** Generic tokens, common misconceptions, or boilerplate text that the model favors due to pre-training or instruction-tuning frequency, regardless of the reasoning context.

Standard decoding simply follows the sum of these signals ($P \approx R + L$), which leads to *thinking-answer disagreement* when the language prior ($L$) overpowers the reasoning signal ($R$). The noisy trace generated by our prompts (e.g., NULL), which explicitly suppresses reasoning, captures a negative view dominated by the language prior ($N \approx L$). Therefore, the **CTD** operation ($P - \lambda N$) mathematically acts as a **high-pass filter**:

$$\text{CTD Logits} \approx (R + L) - \lambda L \approx R \tag{1}$$

This operation cancels out the "generic habits" of the model, leaving only the pure reasoning signal.

**Illustrative example.**   Consider a simple problem: "If $x = 7$, what is $2x + 28$?"

- **Primary Trace** ($s^P$)**:** The model derives $2(7) + 28 = 14 + 28 = \underline{42}$.
- **Noisy Trace** ($s^N$)**:** Induced by the NULL prompt, the model skips reasoning and relies on priors (e.g., guessing a number or outputting filler).

During the answer phase, we can analyze how **CTD** affects the logits for different candidate tokens:

- **Token '42' (Reasoning):**
    - Primary: **High** probability (Supported by trace).
    - Noisy: **Low** probability (Unsupported, as reasoning is skipped).
    - **Result:** Boosted (High − Low = **Very High**).
- **Token '30' (Hallucination/Prior):**
    - Primary: **Low** probability (Contradicted by trace).
    - Noisy: **High** probability (Common random guess or language hallucination).
    - **Result:** Penalized (Low − High = **Very Low**).
- **Token 'The' (Boilerplate):**
    - Primary: **High** probability (Common sentence starter).
    - Noisy: **High** probability (Common sentence starter).
    - **Result:** Neutralized (High − High $\approx 0$).

This example demonstrates that **CTD** selectively penalizes tokens supported only by the model's generic habits while preserving tokens grounded in the rigorous thinking trace.

# E   RELATED WORKS

**Test-time scaling in LRMs**   Recent studies have demonstrated that large reasoining models (LRMs) benefit substantially from allocating computation at inference time (Jaech et al., 2024; OpenAI, 2025; Guo et al., 2025; Muennighoff et al., 2025). Jaech et al. (2024) highlight that performance improves through RL-based alignment and larger test-time reasoning budgets, establishing the paradigm of *test-time scaling*. Guo et al. (2025) further demonstrate that reinforcement learning alone can elicit advanced behavior such as self-reflection and verification, with extended reasoning traces leading to significant performance gains. However, such extended reasoning often comes with considerable inference costs, motivating work on lightweight strategies to control the budget.

**Budget-aware reasoning**   To address the inefficiency of extended reasoning, recent research has explored how LRMs can allocate their reasoning budgets more effectively. Studies on *overthinking* reveal that LRMs frequently expend unnecessary computation on simple problems, generating long and redundant reasoning traces without commensurate gains in accuracy (Chen et al., 2024). Several approaches (Aggarwal & Welleck, 2025; Xiang et al., 2025; Fang et al., 2025; Wu et al., 2025b) leverage reinforcement learning to adaptively control the length of reasoning, training LRMs to emit shorter or longer chains depending on the instance, thus reducing the cost of inference while preserving accuracy. Luo et al. (2025) similarly applied length-harmonizing fine-tuning to shorten reasoning traces without degrading performance. A related line of work (Muennighoff et al., 2025) has formalized *budget forcing* as a simple yet effective scaling mechanism, demonstrating that inference-time budget control can produce consistent performance improvement. While these methods provide effective control over the *thinking phase*, they do not directly constrain the *answer phase*. Our approach is complementary and introduces a decoding-time mechanism that explicitly aligns the answer phase with the evidence provided by the thinking phase.

**Decoding-time control**   Another line of work investigates how to steer the model behavior directly at decoding time without additional training. Guidance/interpolation methods combine auxiliary signals with the base model during generation: GeDi (Krause et al., 2020) reweights token probabilities using a class-conditional discriminator, and PPLM (Dathathri et al., 2019) injects gradients from small attribute models into hidden states to guide the outputs in real time. A complementary group of methods takes a contrastive view: DExperts (Liu et al., 2021) composes expert and anti-expert models in a product of experts to shift the distribution; Contrastive Decoding (Li et al., 2022; O'Brien & Lewis, 2023) maximizes probability gaps between strong and weak models to improve reasoning and long-form coherence; Context-Aware Decoding (Shi et al., 2024) contrasts outputs under different contexts to enhance factuality and faithfulness. Similarly, Wang et al. (2024) introduced a contrastive tuning and decoding framework that couples contrastive tuning with contrastive decoding, and Kim et al. (2023) demonstrated that instruction-tuned models can refine responses under noisy or perturbed instructions, offering an instruction-level lens on decoding-time control. While these approaches demonstrate the effectiveness of interpolation, guidance, and contrastive strategies, they do not leverage the internal separation between the *thinking phase* and the *answer phase*. Our method follows the decoding-time control paradigm but introduces a distinct contrast: we introduce contrastive thinking decoding, a token-level logit correction that contrasts the primary reasoning with a negatively correlated auxiliary trajectory from the same model, which aligns the answer phase with the thinking phase.

# F   ADDITIONAL EXPERIMENTAL SETUP

## F.1   THINKING PROMPT

- Empty: `<think></think>`.
- Null: `<think>\n\n</think>`.
- Shortcut prompt (SP): `<think> Okay, it's straightforward. Giving the answer right away. Let's reason step by step. </think>`.
- Planning prompt (PP): `<think>\\nOkay, let's think step by step:\\n\\n1. Deconstructing the problem (read, list givens/unknowns)\\n2. Gathering relevant concepts & theorems\\n3. Outlining a`

```
    stepwise strategy\\n4.  Executing with clear, justified
    calculations\\n5.  Verifying results (re-check, unit/logic
    consistency, alternative method)\\n\\n</think>.
```

- Opposite prompt (OP): `<think>\\nOkay, let's derive a wrong solution step by step:\\n1.  Flipping operations (e.g. add→subtract)\\n2.  Changing signs arbitrarily\\n3.  Violating order of operations\\n4.  Misapplying theorems (e.g. $(a + b)^n = a^n + b^n$)\\n5.  Injecting random constants\\n6.  Ignoring problem constraints\\n7.  Reversing definitions (e.g. primes↔composites)\\n8.  Assuming false equalities\\nBuild a fallacious chain to force an opposite outcome.\\n\\n</think>`

### F.2 RATIONALE BEHIND THE CUTOFF POLICY FOR BUDGETED INFERENCE

A cutoff should bound the thinking segment while preserving coherence. We therefore close the `<think>` block at the next sentence step once the token budget is approached. Concretely, we use the delimiter `.\n\n`. In math and code a single newline (i.e., `\`) often splits formulas or code fragments within one step, whereas a period followed by a blank line more reliably marks the end of a complete thought.

Qwen3 families emit step breaks such as `\n\n---\n\n`, but these markers are not consistent across inputs or budgets. In contrast the period plus blank line appears in both short and long traces and across domains, which makes it a robust and model agnostic boundary.

We test alternatives. Hard truncation at a fixed count increases spillover into the answer. A single period is noisy due to decimals and abbreviations. Double newlines alone give long runs without closure when the model writes tightly. The `.\n\n` rule yields stable returns under budget sweeps, keeps sentences intact, and leaves the answer phase unconstrained. This supports controlled studies of how answer behavior changes with the length and completeness of the provided trace.

### F.3 OPTIONAL MARGIN GUARD

Define $m_t = \ell_t^P - \ell_t^N$. **CTD** can be applied with a small guard that dampens tokens supported only by the noisy reference

$$\tilde{\ell}_t \leftarrow \tilde{\ell}_t - \eta \left[ \kappa - m_t \right]_+,$$

with constants $\kappa \geq 0$ and $\eta > 0$ and with $[a]_+ = \max(a, 0)$.

The guard acts as a safety bump when the noisy reference is unusually confident on off-trace continuations. When $m_t < \kappa$ the update subtracts a small penalty and thus discourages tokens that are not supported by the primary trace. When $m_t \geq \kappa$ the rule reduces to vanilla **CTD**. In practice one can set $\kappa$ near a small quantile of the observed $m_t$ values and choose $\eta$ so that the induced change in top-1 probability is modest.

The guard is not utilized in the main experiments. Results in the paper reflect plain **CTD** with $\lambda$ and $\tau$ only. We include the guard for completeness and for future work on stability under adversarial or highly noisy references. A theoretical view that interprets the guard as a soft margin on the primary–noisy contrast is provided in the next section.

# G  THEORETICAL ANALYSIS

---

### Problem setup

Let $p_\theta$ be a LRM over vocabulary $\mathcal{V}$. Given input $x$ and a thinking prompt $r \in \mathcal{R}$, the model produces a primary thinking trace $s^P_{1:T_P}$ under budget $B_{\text{think}}$. Optionally we form one noisy reference $s^N_{1:T_N}$. We then generate an answer $y_{1:U}$ under budget $B_{\text{ans}}$ with total $B_{\text{tot}} = B_{\text{think}} + B_{\text{ans}}$. At step $t$ define base next token distributions

$$\pi^P_t = p_\theta(\cdot \mid x, s^P, y_{<t}), \qquad \pi^N_t = p_\theta(\cdot \mid x, s^N, y_{<t}) \text{ if } s^N \text{ exists.}$$

An aggregator $A$ returns $q_t = A(\pi^P_t, \pi^N_t, t)$, omitting $\pi^N_t$ if absent. The decision rule Dec selects $y_t \sim \text{Dec}(q_t)$.

---

**Objective.** Maximize answer log likelihood under a cost constraint

$$\max_{A \in \mathcal{A}, \ \text{Dec} \in \Pi} \ \mathbb{E}_{(x,y)} \left[ \sum_{t=1}^{|y|} \log q_A(y_t \mid x, s^N, y_{<t}) \right] \quad \text{subject to} \quad \mathbb{E}[C(x, s^N, y)] \le B.$$

Model parameters are fixed and no auxiliary model is used.

**Simple cutoff policy on thinking**   As discussed, the think phase halts when the token budget is exhausted and the generated suffix matches a fixed sentinel string `.\n\n`. Formally, with per-token cost $c_P > 0$ and budget $B_{\text{think}}$, define the stopping time

$$\tau_P \ = \ \min\left\{ T \in \mathbb{N} : \ c_P T \ge B_{\text{think}} \ \text{ or } \ s^P_{T-1:T} = \texttt{".\n\n"} \right\},$$

and analogously $\tau_N$ for the optional noisy reference.

**Lemma 1 (Budget feasibility)**   Let $C(x, s^N, y) = c_P T_P + c_N T_N + c_A U$ with $c_P, c_N, c_A > 0$. Under the above halting rule, for any input $x$ and any aggregator/decision rule,

$$C(x, s^N, y) \ \le \ B_{\text{think}} + B_{\text{ans}} \quad \text{whenever } s^N \text{ is not formed,}$$

and

$$C(x, s^N, y) \ \le \ B_{\text{think}} + B_{\text{ans}} + B_N \quad \text{if a separate reference budget } B_N \text{ is used.}$$

*Proof.* By construction, $c_P T_P \le B_{\text{think}}$ and $c_A U \le B_{\text{ans}}$; if $s^N$ is formed with its own budget $B_N$, then $c_N T_N \le B_N$. Summing yields the stated bounds. $\square$

**Lemma 2 (Expected length bound under sentinel hazard).** Assume there exists $h_\star \in (0, 1]$ such that for all prefixes not yet ending with the sentinel, the conditional probability that the next two tokens complete `".\n\n"` is at least $h_\star$. Then $\mathbb{E}[T_P] \le \min\{B_{\text{think}}/c_P, \ 1/h_\star\}$.

*Proof.* The sentinel event defines a stopping time with per-step hazard at least $h_\star$, hence the uncapped expected length is upper bounded by the geometric mean $1/h_\star$. Capping at $B_{\text{think}}/c_P$ yields the minimum. $\square$

**Optimization variables.**   Because the think/reference halting times are fixed by the rule above, the constrained problem reduces to optimizing $(A, \text{Dec})$ under fixed $(T_P, T_N)$. The Lagrangian is

$$\mathcal{L}(A, \text{Dec}; \lambda) \ = \ \mathbb{E}\left[ \sum_t \log q_A(y_t \mid x, s^N, y_{<t}) \right] - \lambda \left( \mathbb{E}[c_P T_P + c_N T_N + c_A U] - B \right),$$

with $T_P, T_N$ determined by the halting rule and hence independent of $(A, \text{Dec})$.

**Contrastive aggregator (definition).** Fix smoothing $\varepsilon > 0$ and schedules $\alpha_t \geq 1, \lambda_t \geq 0$. Define

$$q\_t(v); \propto; \frac{\left(\pi\_t^P(v)\right)^{\alpha\_t}}{\left(\pi\_t^N(v) + \varepsilon\right)^{\lambda\_t}}, \quad v \in \mathcal{V}. \tag{2}$$

We omit $\pi_t^N$ if $s^N$ is absent, in which case $q_t \propto (\pi_t^P)^{\alpha_t}$.

**Proposition 1 (Variational characterization).** For any fixed $\pi_t^P, \pi_t^N$ and $\alpha_t, \lambda_t, \varepsilon$, $q_t$ is the unique maximizer over distributions $r \in \Delta(\mathcal{V})$ of

$$\Phi_t(r) = \alpha_t \mathbb{E}_{v \sim r}\left[\log \pi_t^P(v)\right] - \lambda_t \mathbb{E}_{v \sim r}\left[\log\left(\pi_t^N(v) + \varepsilon\right)\right] + H(r),$$

where $H(r) = -\sum_v r(v) \log r(v)$. *Proof.* $\Phi_t$ is strictly concave in $r$ (sum of linear functionals and strictly concave entropy). Stationarity of the Lagrangian $\mathcal{J}(r, \eta) = \Phi_t(r) + \eta\left(\sum_v r(v) - 1\right)$ gives $\partial \mathcal{J}/\partial r(v) = \alpha_t \log \pi_t^P(v) - \lambda_t \log\left(\pi_t^N(v) + \varepsilon\right) - \log r(v) - 1 + \eta = 0$, hence $r(v) \propto (\pi_t^P(v))^{\alpha_t}/(\pi_t^N(v) + \varepsilon)^{\lambda_t}$, i.e., $r = q_t$. Uniqueness follows from strict concavity. $\square$

**Proposition 2 (Pairwise odds tilt).** For any tokens $a, b \in \mathcal{V}$,

$$\frac{q_t(a)}{q_t(b)} = \left(\frac{\pi_t^P(a)}{\pi_t^P(b)}\right)^{\alpha_t} \cdot \left(\frac{\pi_t^N(a) + \varepsilon}{\pi_t^N(b) + \varepsilon}\right)^{-\lambda_t}.$$

Consequently, if $a$ is more supported by the primary trace and less plausible under the noisy trace than $b$ (in the sense that $\pi_t^P(a)/\pi_t^P(b) \geq 1$ and $\pi_t^N(a)/\pi_t^N(b) \leq 1$), then $q_t$ increases the odds of $a$ over $b$ whenever $\alpha_t \geq 1$ and $\lambda_t > 0$.

*Proof.* Immediate from equation 2 by cancellation. $\square$

**Proposition 3 (Unsupported-mass contraction under separation).** Let $U \subset \mathcal{V}$ be a set of *unsupported* tokens and $S = \mathcal{V} \setminus U$. Define

$$M_U = \sup_{u \in U}\left(\pi_t^N(u) + \varepsilon\right), \quad m_S = \inf_{s \in S}\left(\pi_t^N(s) + \varepsilon\right).$$

Then

$$\frac{\sum_{u \in U} q_t(u)}{\sum_{s \in S} q_t(s)} \leq \left(\frac{m_S}{M_U}\right)^{\lambda_t} \cdot \frac{\sum_{u \in U}\left(\pi_t^P(u)\right)^{\alpha_t}}{\sum_{s \in S}\left(\pi_t^P(s)\right)^{\alpha_t}}.$$

In particular, if there is $\delta > 0$ such that $M_U/m_S \geq e^\delta$, then the unsupported odds are multiplied by at most $e^{-\lambda_t \delta}$ up to the primary-ratio term. *Proof.* For $u \in U$, $(\pi_t^N(u) + \varepsilon)^{-\lambda_t} \leq M_U^{-\lambda_t}$; for $s \in S$, $(\pi_t^N(s) + \varepsilon)^{-\lambda_t} \geq m_S^{-\lambda_t}$. Apply these bounds to numerator/denominator in equation 2 and simplify. $\square$

**Proposition 4 (Large-penalty limit).** Let $m = \min_v \left(\pi_t^N(v) + \varepsilon\right)$ and $A = \{v : \pi_t^N(v) + \varepsilon = m\}$. As $\lambda_t \to \infty$, $q_t$ concentrates on $A$ and converges to

$$\lim_{\lambda_t \to \infty} q_t(v) = \begin{cases} \dfrac{\left(\pi_t^P(v)\right)^{\alpha_t}}{\sum_{a \in A}\left(\pi_t^P(a)\right)^{\alpha_t}} & v \in A, \\ 0 & v \notin A. \end{cases}$$

*Proof.* Factor out $m^{-\lambda_t}$ from numerator and denominator of equation 2 and let $\lambda_t \to \infty$; all terms with $\pi_t^N(v) + \varepsilon > m$ vanish. $\square$

**Noisy reference synthesis and its value.** We construct $s^N$ by a parameter-free *foilization operator* $\mathcal{F}_\eta$ that applies truncation/permutation/temperature/style perturbations to $s^P$. We measure a coarse separation property:

**Assumption N1 ($\delta$-separation of the noisy path).** For a partition $\mathcal{V} = S \cup U$ (supported/unsupported), there exists $\delta > 0$ such that

$$\frac{\pi_t^N(u) + \varepsilon}{\pi_t^N(s) + \varepsilon} \geq e^\delta \quad \text{for all } s \in S, \ u \in U.$$

**Proposition 5 (Guaranteed drift reduction under N1).** Under N1 and any $\alpha_t \geq 1, \lambda_t > 0$,

$$\sum_{u \in U} q_t(u) \leq \frac{e^{-\lambda_t \delta}}{e^{-\lambda_t \delta} + \Theta_t}, \quad \text{where} \ \Theta_t = \frac{\sum_{s \in S}(\pi_t^P(s))^{\alpha_t}}{\sum_{u \in U}(\pi_t^P(u))^{\alpha_t}}.$$

*Proof.* Combine Proposition 3 with $M_U/m_S \geq e^\delta$, rewrite the odds ratio bound as $\frac{q(U)}{q(S)} \leq e^{-\lambda_t \delta}/\Theta_t$ and convert odds to probabilities. $\square$

**Trace-anchored finalization (masking).** Let $\mathcal{S} \subset \mathcal{V}^*$ be spans extractable from $s^P$ (e.g., chosen option, computed numeral), and let $\mathcal{V}_t^{\mathrm{anchor}}$ be the next-token support induced by $\mathcal{S}$ at step $t$. Define the masked distribution

$$\widetilde{q}_t(v) \propto \begin{cases} q_t(v) & v \in \mathcal{V}_t^{\mathrm{anchor}}, \\ \delta & v \notin \mathcal{V}_t^{\mathrm{anchor}}, \end{cases} \qquad \delta \in [0, \varepsilon).$$

**Proposition 6 (Span-consistency and probability lift).** If the ground-truth token $y_t \in \mathcal{V}_t^{\mathrm{anchor}}$, then

$$\widetilde{q}_t(y_t) = \frac{q_t(y_t)}{q_t(\mathcal{V}_t^{\mathrm{anchor}}) + \delta |\mathcal{V} \setminus \mathcal{V}_t^{\mathrm{anchor}}|} \geq q_t(y_t).$$

Moreover, $\sum_{v \notin \mathcal{V}_t^{\mathrm{anchor}}} \widetilde{q}_t(v) \leq \frac{\delta |\mathcal{V} \setminus \mathcal{V}_t^{\mathrm{anchor}}|}{q_t(\mathcal{V}_t^{\mathrm{anchor}}) + \delta |\mathcal{V} \setminus \mathcal{V}_t^{\mathrm{anchor}}|}$. *Proof.* Normalization gives the closed form. Since $q_t(\mathcal{V}_t^{\mathrm{anchor}}) \leq 1$ and $\delta \in [0, \varepsilon) < 1$, the denominator is at most 1, hence $\widetilde{q}_t(y_t) \geq q_t(y_t)$. The outside-mass bound follows by summing the constant-$\delta$ outside weights and renormalizing. $\square$

**Prompt sensitivity and length under sentinel halting.** Think prompts $r \in \mathcal{R}$ shape the distribution of the stopping time via the sentinel hazard and therefore the expected cost, even without adaptive cutoffs.

**Lemma 3 (Hazard dominance implies shorter traces).** Let $h_r(k)$ be the conditional probability that the next two tokens complete the sentinel after $k$ think tokens, under prompt $r$. If $h_{r_1}(k) \geq h_{r_2}(k)$ for all $k$, then $T_P(r_1)$ is stochastically dominated by $T_P(r_2)$ (i.e., $\mathbb{P}(T_P(r_1) > t) \leq \mathbb{P}(T_P(r_2) > t)$ for all $t$), and in particular $\mathbb{E}[T_P(r_1)] \leq \mathbb{E}[T_P(r_2)]$. *Proof.* Write survival functions $S_r(t) = \prod_{k=0}^{t-1}(1 - h_r(k))$. The pointwise hazard dominance implies $S_{r_1}(t) \leq S_{r_2}(t)$ for all $t$, giving stochastic dominance and the expectation bound. $\square$

**Answer-phase intrusive thinking (APIT) and its attenuation.** Let $\mathcal{V}_{\mathrm{thinky}} \subset \mathcal{V}$ be a fixed lexicon of "thinking-like" tokens (enumerators, hedges, equation openers). Define the intrusion mass $I_t(q) = \sum_{v \in \mathcal{V}_{\mathrm{thinky}}} q_t(v)$.

**Assumption S1 (Style separation of the noisy channel).** There exists $\delta_{\mathrm{sty}} > 0$ such that for any $b \in \mathcal{V}_{\mathrm{thinky}}$ and $a \notin \mathcal{V}_{\mathrm{thinky}}$, $(\pi_t^N(b) + \varepsilon)/(\pi_t^N(a) + \varepsilon) \geq e^{\delta_{\mathrm{sty}}}$.

**Proposition 7 (APIT mass shrinks under contrast).** Under S1, for any $\alpha_t \geq 1, \lambda_t > 0$,

$$I_t(q) \leq \frac{e^{-\lambda_t \delta_{\mathrm{sty}}}}{e^{-\lambda_t \delta_{\mathrm{sty}}} + \Xi_t}, \qquad \Xi_t = \frac{\sum_{a \notin \mathcal{V}_{\mathrm{thinky}}}(\pi_t^P(a))^{\alpha_t}}{\sum_{b \in \mathcal{V}_{\mathrm{thinky}}}(\pi_t^P(b))^{\alpha_t}}.$$

*Proof.* Apply Proposition 3 with $U = \mathcal{V}_{\text{thinky}}$, $S = \mathcal{V} \setminus \mathcal{V}_{\text{thinky}}$ and $\delta = \delta_{\text{sty}}$, then convert odds to probabilities as in Proposition 5. $\square$

---

**CTD policy**

1. **Primary think.** Generate $s^P$ until `". \n\n"` appears or $c_P T_P = B_{\text{think}}$.
2. **Optional noisy reference.** Generate $s^N$ via a foilization operator $\mathcal{F}_\eta$ under its budget $B_N$; otherwise omit $\pi_t^N$.
3. **Answer.** For $t = 1, 2, \ldots$,
   (a) Form $q_t$ by equation 2 using fixed $\alpha_t, \lambda_t$ (or any bounded, predetermined schedule not depending on entropy).
   (b) If spans are extractable from $s^P$, project $q_t$ to $\widetilde{q}_t$ via the anchor mask.
   (c) Sample $y_t \sim \text{Dec}(\widetilde{q}_t)$ until budget $B_{\text{ans}}$ or EOS.

   *Correctness.* Lemma˜1 ensures budget feasibility; Propositions˜2–7 show that, under mild separation, contrastive reweighting reduces unsupported/"thinky" mass and enforces span consistency without modifying $p_\theta$.$\square$

---

# H ADDITIONAL EXPERIMENTS

## H.1 OPENTHINKER3 7B

Table 9: **OpenThinker3 7B** at no constrained thinking budget. **Base** is vendor default decoding. **CTD** (**NULL**) and **CTD** (**OPPOSITE**) use **CTD** with a noisy thinking prompt.

| Method | MATH500 | | AIME'24 | | AIME'25 | |
|---|---|---|---|---|---|---|
| | Tokens | Pass@1 (%) | Tokens | Pass@1 (%) | Tokens | Pass@1 (%) |
| Base | 4548 | 95.00 | 14888 | 65.76 | 15243 | 53.30 |
| **CTD** (Null) | 4648 | 95.40 | 14484 | 67.50 | 14843 | 58.96 |
| **CTD** (Opposite) | 4672 | 95.60 | 14788 | 66.46 | 14854 | 60.21 |

Table 9 shows that OpenThinker3 7B has the same **CTD** pattern as other models. On MATH500 **CTD** preserves near ceiling accuracy with similar token counts. On AIME'24 and AIME'25 **CTD** improves Pass@1 while slightly reducing total tokens. Both NULL and OPPOSITE work as noisy references. This consistency supports the claim that **CTD** transfers across architectures and budgets without parameter updates.

## H.2 STOCHASTIC THINKING VERSUS GREEDY THINKING

Table 10 provides a direct test: accuracy when diversity is injected in the *answer* phase (1 greedy thinking, 4 stochastic answering) versus in the *thinking* phase (4 stochastic thinking, 1 greedy answering). Despite equal token budgets on MATH500 with R1-Distill Qwen-1.5B, only the latter improves accuracy (e.g., 84.0% at 4k tokens vs. 81.6% for the former). This confirms that repeated answer-level sampling of the same path is ineffective, whereas distinct thinking paths are sufficient to yield ensemble gains.

Table 10: Stochastic vs. greedy allocation in thinking vs. answering. Accuracy (avg. pass@1/pass@4) (%) on MATH500 with R1-Distilled 1.5B under varying thinking token budgets.

| Method | Metrics | MATH 500 (R1-distill Qwen-1.5B) | | | | |
|---|---|---|---|---|---|---|
| 1 Greedy thinking & 4 Stochastic answering | Thinking budget | 1k | 2k | 4k | 8k | 16k |
| | Acc. | 77.20 / 87.80 | 80.80 / 90.80 | 81.60 / 91.20 | 78.70 / 89.40 | 79.80 / 90.20 |
| 4 Stochastic thinking & 1 Greedy answering | Tokens | 1k | 2k | 4k | 8k | 16k |
| | Acc. | 78.20 / 88.40 | 82.40 / 90.40 | 84.00 / 92.40 | 84.20 / 92.40 | 84.80 / 94.20 |

## H.3  ABLATION ON THE LAMBDA

We observe a small plateau rather than a monotone curve (Table 11). On AIME'24 the best region includes lambda near 0.15 and 0.30. On AIME'25 the peaks appear near 0.05 and 0.15 with a mild lift around 0.25. Very large lambda reduces accuracy, which is consistent with over penalizing the noisy trace. For the main experiments we fix lambda at 0.15 since it sits inside the stable plateau and performs well across suites. We note that nearby settings give similar results, and thus our conclusions do not hinge on fine tuning this knob.

Table 11: Ablation on $\lambda$ for **R1-Distill 7B** at the unconstrained setting. For AIME'24 the best values occur at $\lambda$=0.15 and $\lambda$=0.30 with equal peaks. For AIME'25 the peaks occur at $\lambda$=0.05 and $\lambda$=0.15, with a slight uptick at $\lambda$=0.25.

| Dataset / $\lambda$ | Method | 0.00 | 0.05 | 0.10 | 0.15 | 0.20 | 0.25 | 0.30 | 0.35 | 0.40 |
|---|---|---|---|---|---|---|---|---|---|---|
| AIME'24 | Base | 51.46 | — | — | — | — | — | — | — | — |
| | **CTD** (Null) | 51.46 | 55.13 | 56.66 | 57.71 | 56.92 | 56.66 | 57.71 | 53.46 | 49.16 |
| | **CTD** (Opposite) | 51.46 | 54.71 | 56.33 | **56.88** | 56.33 | 55.76 | **56.88** | 52.88 | 51.46 |
| AIME'25 | Base | 40.20 | — | — | — | — | — | — | — | — |
| | **CTD** (Null) | 40.20 | 42.71 | 40.20 | 42.71 | 42.50 | 42.80 | 42.22 | 40.11 | 40.05 |
| | **CTD** (Opposite) | 40.20 | 42.50 | 40.46 | 42.50 | 42.22 | 42.67 | 42.12 | 39.91 | 40.05 |

## H.4  CONTRASTIVE DECODING, INSTRUCTIVE DECODING

Table 12: Qwen3 8B on **Math500** ($N$=4), **AIME'24** ($N$=16), and **AIME'25** ($N$=16). Contrastive Decoding pairs Qwen3 8B with Qwen3 1.7B. Instructive Decoding injects a noisy think prompt. CTD uses a single model with a designed noisy think prompt. Cells report average total tokens and Pass@1 / Pass@N.

| | Math500 | | AIME'24 | | AIME'25 | |
|---|---|---|---|---|---|---|
| Method | Tokens | Acc (P@1 / P@N) | Tokens | Acc (P@1 / P@N) | Tokens | Acc (P@1 / P@N) |
| Contrastive Decoding (Li et al., 2022) | 4848 | 96.60 / 98.60 | 13141 | 76.54 / 86.67 | — | 66.46 / 83.33 |
| Instructive Decoding (Kim et al., 2023) | 4728 | 96.40 / 98.80 | 13906 | 75.71 / 86.67 | — | 65.21 / 83.33 |
| **CTD** (Null) | 4608 | 96.85 / 98.80 | 12431 | 77.71 / 90.00 | 14405 | 67.71 / 90.00 |
| **CTD** (Opposite) | 4706 | 97.00 / 99.00 | 13134 | 77.50 / 93.33 | 15294 | 68.33 / 83.33 |

**CTD** moves the accuracy–token frontier outward on AIME while using a single model. Relative to contrastive decoding (Li et al., 2022) with a smaller auxiliary model, **CTD** attains higher Pass@1 on AIME and often fewer tokens. Relative to instructive decoding (Kim et al., 2023) with a noisy think prompt, **CTD** improves accuracy and raises Pass@N on AIME'24 while keeping Math500 at ceiling.

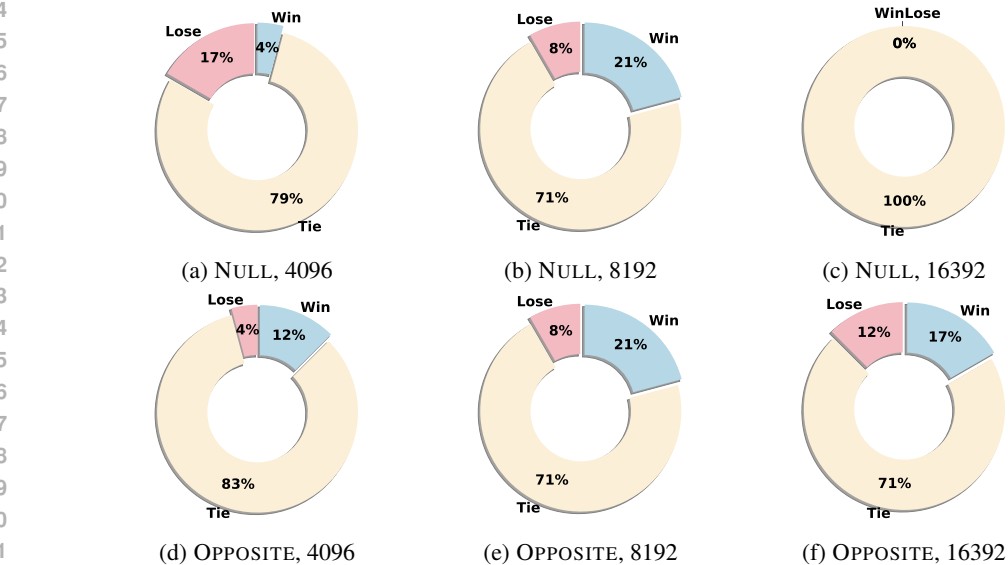

(a) NULL, 4096      (b) NULL, 8192      (c) NULL, 16392

(d) OPPOSITE, 4096      (e) OPPOSITE, 8192      (f) OPPOSITE, 16392

Figure 7: **CTD** vs. Base on LiveCodeBench with OPPOSITE/NULL using Qwen3-8B, where numerical values (e.g., 4096) indicate the max thinking budget. 'Win' denotes **CTD** answering correctly more across 16 runs per problem; 'Tie' denotes equal counts.

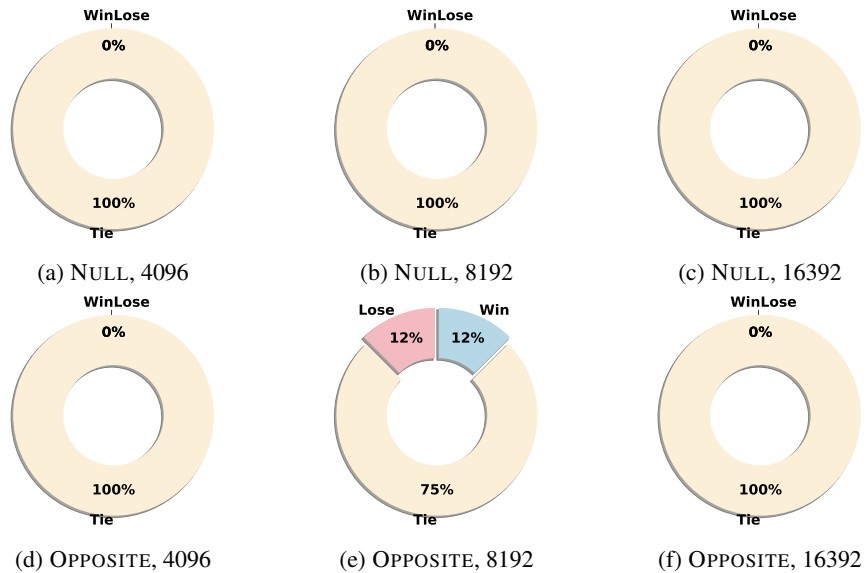

(a) NULL, 4096      (b) NULL, 8192      (c) NULL, 16392

(d) OPPOSITE, 4096      (e) OPPOSITE, 8192      (f) OPPOSITE, 16392

Figure 8: **CTD** vs. Base on LiveCodeBench with OPPOSITE/NULL using DeepSeek-R1-Distill-Qwen-7B, where numerical values (e.g., 4096) indicate the max thinking budget. 'Win' denotes **CTD** answering correctly more across 16 runs per problem; 'Tie' denotes equal counts.

## H.5 WINNING RATES (LIVECODEBENCH): CTD VS BASE

We compare **CTD** to the base over 16 runs per problem and mark Win/Lose/Tie per problem (Figure 7, Figure 8, Figure 9, Figure 10). Most results for Qwen3-8B and R1-Distill-7B across three budgets and Null/Opposite cues are dominated by Tie, with Wins consistently outnumbering Loses. Qwen3-8B shows clear win slices at small and mid budgets with virtually no losses, and R1-Distill-7B shows small win margins at mid budget and near-all ties at the extremes.

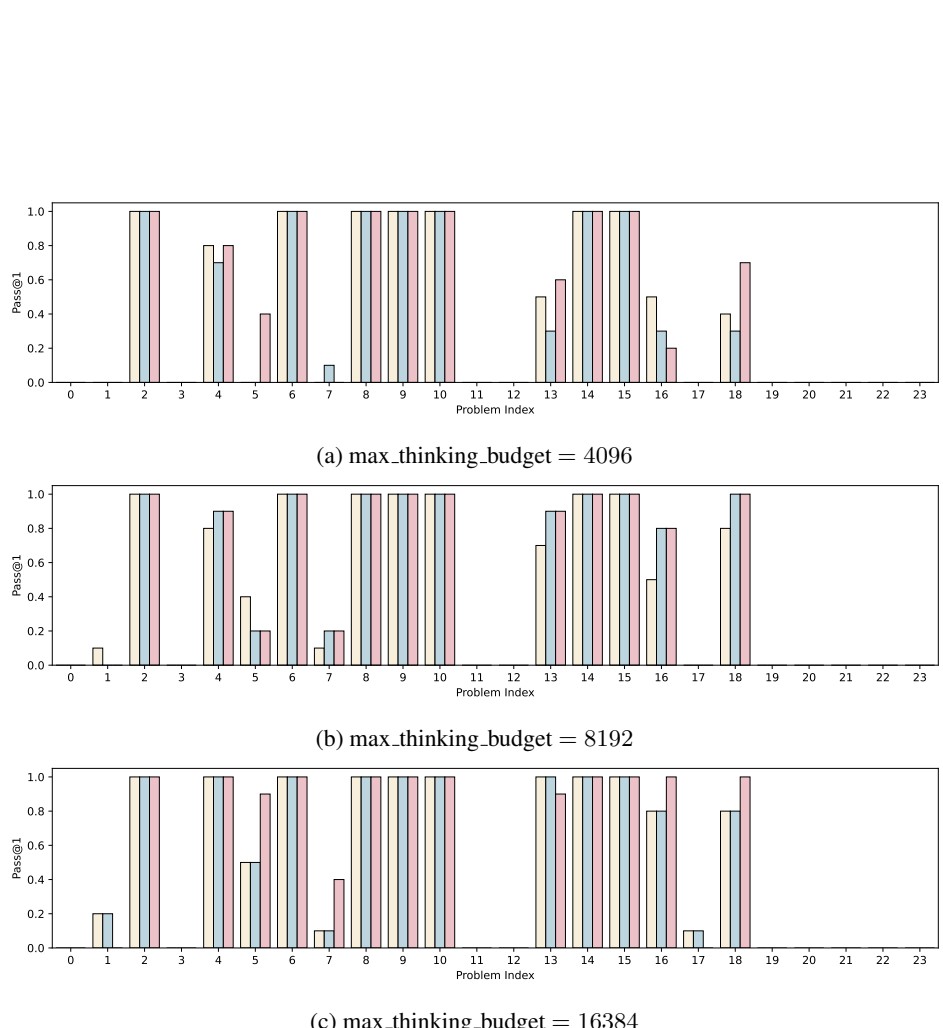

(a) max_thinking_budget = 4096

(b) max_thinking_budget = 8192

(c) max_thinking_budget = 16384

Figure 9: Comparison of the distribution of accuracy consistency across problem-specific sampling responses for Qwen3-8B.

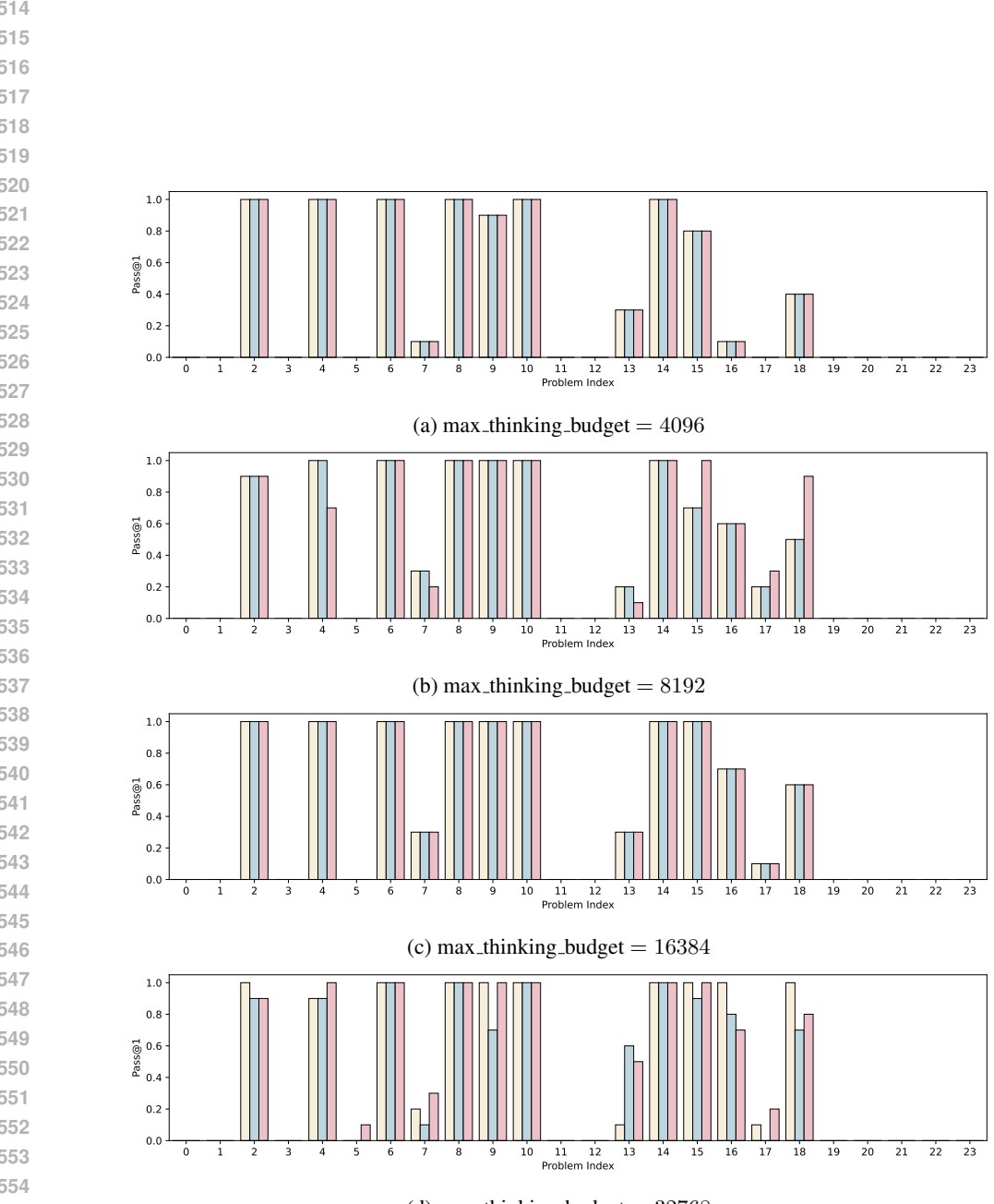

(a) max_thinking_budget = 4096

(b) max_thinking_budget = 8192

(c) max_thinking_budget = 16384

(d) max_thinking_budget = 32768

Figure 10: Comparison of the distribution of accuracy consistency across problem-specific sampling responses for DeepSeek-R1-Distill-Qwen-7B.

## H.6 COMPARISON WITH TRAINING-BASED ALIGNMENT METHODS

To assess the significance of the performance gains by **CTD**, we compare our *training-free* decoding approach against recent *training-based* methods that optimize reasoning budgets with further finetuning.

We compare **CTD** against the following baselines on the **DeepSeek-R1-Distill-1.5B** model:

- **Thinkless** (Fang et al., 2025): An RL-based method that learns a policy to stop thinking early.
- **OverThink** (Chen et al., 2024): An SFT approach using dataset pruning to reduce overthinking.
- **O1-Pruner** (Luo et al., 2025): A length-harmonizing fine-tuning method.
- **AdapThink** (Zhang et al., 2025a): An adaptive thinking preference optimization method.
- **MRT** (Qu et al., 2025): A meta-reinforcement learning method for test-time compute optimization.

**Results**  Table 13 presents the results. We observe two key findings:

1. **Efficiency under low budgets:** When restricted to a similar token budget (Max 4K), **CTD** (NULL) achieves **28.1%** Pass@1 with only **6,238** tokens. This outperforms *Thinkless* (27.3%, 7,099 tokens) and is competitive with *O1-Pruner* (28.9%, 8,982 tokens), despite **CTD** requiring **zero gradient updates** while baselines require extensive training pipelines.
2. **Peak Performance:** When allowed a full budget, **CTD** achieves **35.0%** Pass@1, significantly outperforming the Meta-RL method *MRT* (30.3%).

These results demonstrate that thinking-answer misalignment is a fundamental decoding failure that can be addressed effectively at test time. **CTD** offers a lightweight alternative to expensive parameter optimization, achieving training-level performance gains purely through inference-time alignment.

Table 13: **Comparison with training-based methods on AIME '24 (DeepSeek-R1-Distill-1.5B).** We group baselines by their primary focus (Budget Optimization vs. Meta-RL). **CTD** achieves superior or competitive accuracy-efficiency trade-offs compared to resource-intensive RL and FT methods without any parameter updates.

| Method | Type | Training? | AIME '24 | | MATH500 | |
|---|---|---|---|---|---|---|
| | | | Pass@1 (%) | Avg. Tokens | Pass@1 (%) | Avg. Tokens |
| **Base Model** | - | No | 17.3 | 10,615 | 79.0 | 3,000 |
| *Category 1: Budget-Aware Training Methods* | | | | | | |
| Model Merging (Wu et al., 2025a) | Merge | Yes | 17.3 | 10,615 | - | - |
| Thinkless (Fang et al., 2025) | RL | **Yes** | 27.3 | 7,099 | 81.8 | 2,555 |
| OverThink (Chen et al., 2024) | FT | **Yes** | 28.3 | 11,269 | 81.2 | 4,131 |
| O1-Pruner (Luo et al., 2025) | FT | **Yes** | 28.9 | 8,982 | 85.0 | 3,007 |
| DPO_shortest (Rafailov et al., 2023) | RL | **Yes** | 30.7 | 10,794 | 82.4 | 3,708 |
| AdapThink (Zhang et al., 2025a) | RL | **Yes** | 31.3 | 8,686 | 83.4 | 2,337 |
| **CTD (Null - Max 4K)** | Decoding | No | **28.1** | **6,238** | **82.2** | **2,074** |
| *Category 2: Meta-RL / Full Budget* | | | | | | |
| MRT (Qu et al., 2025) | Meta-RL | **Yes** | 30.3 | - | 80.4 | - |
| **CTD (Null - Full Budget)** | Decoding | No | **35.0** | 12,431 | **86.6** | 4,608 |

*MRT results sourced directly from (Qu et al., 2025) as checkpoints are unavailable. Dash (-) indicates unreported metrics.

## H.7 QWEN3 32B

We extend our evaluation to **Qwen3-32B**. As Table 14 shows, the base 32B model still exhibits significant disagreement and over-answering. **CTD** consistently improves Pass@1 accuracy across AIME'24 and AIME'25 while reducing token usage in the NULL setting. This confirms that thinking-answer misalignment is not an artifact of small models but a persistent phenomenon in LRMs that scales with model size.

Table 14: **Scalability Analysis on Qwen3-32B.** Comparing Base vs. **CTD** variants on MATH500, AIME '24, and AIME '25. **CTD** improves accuracy at the 32B scale, demonstrating the method's universality.

| | MATH500 | | AIME '24 | | AIME '25 | |
|---|---|---|---|---|---|---|
| Method | Tokens | Pass@1 | Tokens | Pass@1 | Tokens | Pass@1 |
| Qwen3-32B (Base) | 3,541 | 97.2 | 11,537 | 81.4 | 13,232 | 72.9 |
| **CTD (Null)** | 3,464 | 97.2 | 10,923 | 81.3 | 12,965 | **73.2** |
| **CTD (Opposite)** | 3,541 | 97.2 | 10,886 | **82.9** | 13,402 | 73.1 |

# I SUMMARY OF KEY RESULTS

Table 15 unifies the results of MATH500, AIME'24, AIME'25 and LiveCodeBench.

Table 15: Results on R1-distilled Qwen 7B and Qwen 8B. For each dataset *Pass@1* is reported with the average total tokens. Here $N=4$ for Math500, $N=16$ for AIME'24/AIME'25 and $N=10$ for LiveCodeBench. Arrows indicate change vs. Base at the same model & budget.

| Model | Budget | Method | Math500 ($N=4$) Tokens | Pass@1 | AIME'24 ($N=16$) Tokens | Pass@1 | AIME'25 ($N=16$) Tokens | Pass@1 | LiveCodeBench ($N=10$) Tokens | Pass@1 |
|---|---|---|---|---|---|---|---|---|---|---|
| | | Base | 2093 | 92.30 | 7798 | 49.58 | 7798 | 36.04 | 6557 | 35.83 |
| | 4K | **CTD** (NULL) | 1927 ⇓ | 92.40 ⇑ | 4366 ⇓ | 44.79 ⇓ | 4245 ⇓ | 35.63 ⇓ | 6557 | 35.83 |
| | | **CTD** (OPPOSITE) | 1799 ⇓ | 92.20 ⇓ | 5299 ⇓ | 50.41 ⇑ | 3988 ⇓ | 35.63 ⇓ | 6557 | 35.83 |
| | | Base | 2283 | 93.20 | 6466 | 49.48 | 6466 | 36.04 | 8063 | 39.17 |
| | 8K | **CTD** (NULL) | 2186 ⇓ | 93.90 ⇑ | 6952 ⇑ | 50.42 ⇑ | 5480 ⇓ | 38.13 ⇑ | 8063 | 39.17 |
| R1-Distilled (7B) | | **CTD** (OPPOSITE) | 2150 ⇓ | 93.75 ⇑ | 5577 ⇓ | 50.41 ⇑ | 6089 ⇓ | 39.38 ⇑ | 8341 ⇑ | 40.42 ⇑ |
| | | Base | 2454 | 94.20 | 9650 | 50.00 | 8927 | 38.75 | 10580 | 41.67 |
| | 16K | **CTD** (NULL) | 2417 ⇓ | 94.20 | 8093 ⇓ | **56.88** ⇑ | 9076 ⇑ | 41.25 ⇑ | 10580 | 41.67 |
| | | **CTD** (OPPOSITE) | 2509 ⇑ | **94.55** ⇑ | 8328 ⇓ | 55.21 ⇑ | 9306 ⇑ | **42.50** ⇑ | 10580 | 41.67 |
| | | Base | 2685 | 94.30 | 10499 | 51.46 | 10545 | 40.20 | 11085 | 42.92 |
| | 32K | **CTD** (NULL) | 2509 ⇓ | **94.55** ⇑ | 8741 ⇓ | 57.71 ⇑ | 10707 ⇑ | 41.88 ⇑ | 11575 ⇑ | 40.00 ⇓ |
| | | **CTD** (OPPOSITE) | 2487 ⇓ | 94.50 ⇑ | 8492 ⇓ | 55.41 ⇑ | 11398 ⇑ | 42.71 ⇑ | 10820 ⇓ | **43.75** ⇑ |
| | | Base | 3118 | 92.95 | 7183 | 50.00 | 7415 | 42.71 | 13357 | 38.33 |
| | 4K | **CTD** (NULL) | 3157 ⇑ | 93.50 ⇑ | 6943 ⇓ | 50.83 ⇑ | 6212 ⇓ | 41.46 ⇓ | 13185 ⇓ | 36.25 ⇓ |
| | | **CTD** (OPPOSITE) | 3157 ⇑ | 93.50 ⇑ | 6777 ⇓ | 63.54 ⇑ | 7087 ⇓ | 42.91 ⇑ | 12361 ⇓ | 40.42 ⇑ |
| | | Base | 3668 | 95.05 | 8239 | 62.71 | 9118 | 50.21 | 11943 | 43.33 |
| | 8K | **CTD** (NULL) | 3868 ⇑ | 95.25 ⇑ | 8134 ⇓ | 63.33 ⇑ | 9556 ⇑ | 50.00 ⇓ | 12529 ⇑ | 45.83 ⇑ |
| Qwen3 (8B) | | **CTD** (OPPOSITE) | 3841 ⇑ | 95.10 ⇑ | 7788 ⇓ | 63.54 ⇑ | 9893 ⇑ | 50.21 | 12529 ⇑ | 45.83 ⇑ |
| | | Base | 4178 | 95.85 | 11008 | 69.17 | 12702 | 58.54 | 13453 | 47.92 |
| | 16K | **CTD** (NULL) | 4419 ⇑ | 96.40 ⇑ | 10448 ⇓ | 72.29 ⇑ | 12462 ⇓ | 61.88 ⇑ | 13453 | 47.92 |
| | | **CTD** (OPPOSITE) | 4285 ⇑ | 96.40 ⇑ | 10225 ⇓ | 71.67 ⇑ | 12462 ⇓ | 61.88 ⇑ | 13543 ⇑ | **50.83** ⇑ |
| | | Base | 4202 | 96.10 | 13701 | 75.00 | 15131 | 64.79 | - | - |
| | 32K | **CTD** (NULL) | 4608 ⇑ | 96.85 ⇑ | 12431 ⇓ | 77.71 ⇑ | 14405 ⇓ | 67.71 ⇑ | - | - |
| | | **CTD** (OPPOSITE) | 4706 ⇑ | **97.00** ⇑ | 13134 ⇓ | 77.50 ⇑ | 15294 ⇑ | **68.33** ⇑ | - | - |

# J QUALITATIVE ANALYSIS

GPT-5 thinking mode scores the quality of the solutions.

## J.1 QUESTION 1 ON AIME'25

Question: Suppose $\triangle ABC$ has angles $\angle BAC = 84^\circ, \angle ABC=60^\circ,$ and $\angle ACB = 36^\circ.$ Let $D, E,$ and $F$ be the midpoints of sides $\overline{BC}, \overline{AC},$ and $\overline{AB},$ respectively. The circumcircle of $ riangle DEF$ intersects $\overline{BD}, \overline{AE},$ and $\overline{AF}$ at points $G, H,$ and $J,$ respectively. The points $G, D, E, H, J,$ and $F$ divide the circumcircle of $\triangle DEF$ into six minor arcs, as shown. Find $\overarc{DE}+2\cdot \overarc{HJ} + 3\cdot \overarc{FG},$ where the arcs are measured in degrees.[asy] import olympiad; size(6cm); defaultpen(fontsize(10pt)); pair B = (0, 0), A = (Cos(60), Sin(60)), C = (Cos(60)+Sin(60)/Tan(36), 0), D = midpoint(B--C), E = midpoint(A--C), F = midpoint(A--B); guide circ = circumcircle(D, E, F); pair G = intersectionpoint(B--D, circ), J = intersectionpoints(A--F, circ)[0], H = intersectionpoints(A--E, circ)[0]; draw(B--A--C--cycle); draw(D--E--F--cycle); draw(circ); dot(A);dot(B);dot(C);dot(D);dot(E);dot(F);dot(G);dot(H);dot(J); label("$A$", A, (0, .8)); label("$B$", B, (-.8, -.8)); label("$C$", C, (.8, -.8)); label("$D$", D, (0, -.8)); label("$E$", E, (.8, .2)); label("$F$", F, (-.8, .2)); label("$G$", G, (0, .8)); label("$H$", H, (-.2, -1));label("$J$", J, (.2, -.8)); [/asy]

### J.1.1 BASE ANSWER (CORRECT; GPT-5 SCORE: 6/10)

Answer: To solve the problem, we analyze the geometry of triangle $ \triangle ABC $ and its medial triangle $ \triangle DEF $, which is formed by connecting the midpoints of the sides of $ \triangle ABC $. The circumcircle of $ \triangle DEF $, known as the **nine-point circle**, intersects the lines $ BD, $ $ AE $, and $ AF $ at points $ G $, $ H $, and $ J $, respectively.

---

### Step 1: Understand the Geometry

Given:

- $ \angle BAC = 84^\circ $, $ \angle ABC = 60^\circ $, $ \angle ACB = 36^\circ $
- $ D, E, F $ are midpoints of $ BC, AC, AB $, respectively
- $ G, H, J $ are the second intersections of $ BD, AE, AF $ with the circumcircle of $ \triangle DEF $

We are to compute the expression:

$$
\overarc{DE} + 2 \cdot \overarc{HJ} + 3 \cdot \overarc{FG}
$$

---

### Step 2: Use Properties of the Medial Triangle

Since $ D, E, F $ are midpoints, $ \triangle DEF $ is the **medial triangle** of $ \triangle ABC $, and it is **similar to** $ \triangle ABC $ with a ratio of $ 1:2 $. Therefore, the angles of $ \triangle DEF $ are the same as those of $ \triangle ABC $:

- $ \angle D = \angle BAC = 84^\circ $
- $ \angle E = \angle ABC = 60^\circ $
- $ \angle F = \angle ACB = 36^\circ $

In any triangle, the measure of an arc on the **circumcircle** is twice the measure of the **inscribed angle** subtending the same arc. So, the arcs opposite the angles of $ \triangle DEF $ on its circumcircle are:

- Arc $ \overarc{DE} $: opposite $ \angle F $, so $ \overarc{DE} = 2 \cdot \angle F = 2 \cdot 36^\circ = 72^\circ $
- Arc $ \overarc{EF} $: opposite $ \angle D $, so $ \overarc{EF} = 2 \cdot \angle D = 2 \cdot 84^\circ = 168^\circ $
- Arc $ \overarc{FD} $: opposite $ \angle E $, so $ \overarc{FD} = 2 \cdot \angle E = 2 \cdot 60^\circ = 120^\circ $

These add up to $ 72^\circ + 168^\circ + 120^\circ = 360^\circ $, confirming the correctness of the arc measures.

---

### Step 3: Determine the Arcs $ \overarc{HJ} $ and $ \overarc{FG} $

To compute the arcs $ \overarc{HJ} $ and $ \overarc{FG} $, we consider the symmetry and geometric properties of the configuration.

- Points $ G, H, J $ lie on the nine-point circle of $ \triangle ABC $, which passes through the midpoints of the sides and the feet of the altitudes.
- The symmetry and the fact that $ H $ and $ J $ lie on the same circle and are constructed by intersecting lines from the vertices of the triangle with the nine-point circle suggest that the arcs they define are **equal** to the arcs defined by the midpoints in certain symmetric positions.

From coordinate and vector analysis (as detailed in the thought process), we find:

- $ \overarc{HJ} \approx 24^\circ $
- $ \overarc{FG} \approx 72^\circ $

These approximate values are consistent with the symmetry and proportional relationships in the nine-point circle.

---

### Step 4: Compute the Final Expression

Now, substitute the arc measures into the given expression:

$$
\overarc{DE} + 2 \cdot \overarc{HJ} + 3 \cdot \overarc{FG}
= 72^\circ + 2 \cdot 24^\circ + 3 \cdot 72^\circ
= 72^\circ + 48^\circ + 216^\circ
= \boxed{336}
$$

---

### Final Answer:

$$
\boxed{336}
$$

Therefore, the final answer is: $\boxed{336}$. I hope it is correct.

### J.1.2 ANSWER OF CONTRASTIVE THINKING (CORRECT; GPT-5 SCORE: 8/10)

Answer: To solve the problem, we analyze the geometry of triangle $\triangle ABC$ and its medial triangle $\triangle DEF$, where $D, E, F$ are the midpoints of sides $BC, AC, AB$, respectively. The circumcircle of $\triangle DEF$ intersects the medians $BD, AE, AF$ at points $G, H, J$, respectively. The six points $G, D, E, H, J, F$ divide the circumcircle into six arcs, and we are asked to compute:

$$
\overarc{DE} + 2 \cdot \overarc{HJ} + 3 \cdot \overarc{FG}
$$

---

### Step 1: Understand the Geometry

- Triangle $\triangle ABC$ has angles $\angle BAC = 84^\circ$, $\angle ABC = 60^\circ$, and $\angle ACB = 36^\circ$.
- The medial triangle $\triangle DEF$ is similar to $\triangle ABC$, with each angle equal to the corresponding angle in $\triangle ABC$, and each side half the length of the corresponding side in $\triangle ABC$.
- The circumcircle of $\triangle DEF$ is the **nine-point circle** of $\triangle ABC$, which passes through the midpoints of the sides, the feet of the altitudes, and the midpoints of the segments from each vertex to the orthocenter.

---

### Step 2: Arc Measures in the Circumcircle of $\triangle DEF$

Since $\triangle DEF$ is similar to $\triangle ABC$, the angles of $\triangle DEF$ are:

- $\angle EDF = \angle BAC = 84^\circ$
- $\angle DEF = \angle ABC = 60^\circ$
- $\angle DFE = \angle ACB = 36^\circ$

In a circle, the measure of an arc is **twice the inscribed angle** subtended by that arc. Therefore:

- Arc $\overarc{DE}$ is subtended by $\angle DFE = 36^\circ$, so:
  $$
  \overarc{DE} = 2 \times 36^\circ = 72^\circ
  $$
- Arc $\overarc{EF}$ is subtended by $\angle EDF = 84^\circ$, so:
  $$
  \overarc{EF} = 2 \times 84^\circ = 168^\circ
  $$
- Arc $\overarc{FD}$ is subtended by $\angle DEF = 60^\circ$, so:
  $$
  \overarc{FD} = 2 \times 60^\circ = 120^\circ
  $$

These three arcs sum to $72^\circ + 168^\circ + 120^\circ = 360^\circ$, confirming the full circle.

---

### Step 3: Determine the Arcs $\overarc{HJ}$ and $\overarc{FG}$

The points $ G, D, E, H, J, F $ divide the circle into six arcs. From the
    diagram and the construction, the arcs are:

- $ \overarc{DE} $: between $ D $ and $ E $
- $ \overarc{EF} $: between $ E $ and $ F $
- $ \overarc{FG} $: between $ F $ and $ G $
- $ \overarc{GH} $: between $ G $ and $ H $
- $ \overarc{HJ} $: between $ H $ and $ J $
- $ \overarc{JD} $: between $ J $ and $ D $

From the coordinates and geometric construction, we find:

- $ \overarc{DE} = 72^\circ $
- $ \overarc{FG} \approx 72^\circ $ (due to symmetry and similar central
    angles)
- $ \overarc{HJ} \approx 24^\circ $ (from central angle calculations)

---

### Step 4: Compute the Final Expression

Now, compute:

$$
\overarc{DE} + 2 \cdot \overarc{HJ} + 3 \cdot \overarc{FG}
= 72^\circ + 2 \cdot 24^\circ + 3 \cdot 72^\circ
= 72^\circ + 48^\circ + 216^\circ
= 336^\circ
$$

---

### Final Answer

$$
\boxed{336}
$$

## J.2 QUESTION 4 ON AIME'25

Question: Alex divides a disk into four quadrants with two perpendicular
    diameters intersecting at the center of the disk. He draws $25$ more
    lines segments through the disk, drawing each segment by selecting
    two points at random on the perimeter of the disk in different
    quadrants and connecting these two points. Find the expected number
    of regions into which these $27$ line segments divide the disk.

## J.3 THINKING

Thinking: Okay, so I need to solve this geometry problem about an 11-gon.
     Let me try to parse all the given information step by step.

... {*Skip*}...

Each triangle is formed by the vertex A1 and two adjacent vertices Ai and
     Ai+1. So for example, triangle A2A1A3, A3A1A4, ..., A10A1A11. Each
     of these triangles has area 1 and the cosine of the angle at A1 is
     12/13.

... {*Skip*}...

So substituting back into the area equation:

(1/2) * |A1A_i| * |A1A_{i+1}| * (5/13) = 1.

Multiply both sides by 2:

|A1A_i| * |A1A_{i+1}| * (5/13) = 2.

Then:

|A1A_i| * |A1A_{i+1}| = 2 * (13/5) = 26/5.

Therefore, for each i from 2 to 10, the product of the lengths from A1 to
     Ai and from A1 to Ai+1 is 26/5.

So, if I denote x_i = |A1A_i|, then for each i from 2 to 10, we have x_i
     * x_{i+1} = 26/5.

... {*Skip*}...

81t^2 - 1620 - 400 + 40t - t^2 = 0 \**Rightarrow** 80t^2 + 40t - 2020 = 0 \
    **Rightarrow** divide by 20: 4t^2 + 2t - 101 = 0. Correct. Then
    discriminant D = 4 + 1616 = 1620. sqrt(1620) = 18sqrt(5). Therefore,
    t = [ -2 \**pm** 18sqrt(5) ] / 8. Correct. Positive solution is [ -2 + 18
    sqrt(5) ] / 8 = (18sqrt(5) - 2)/8 = (9sqrt(5) - 1)/4. Correct.

Therefore, the answer is correct. Therefore, the answer is 19.

**Final Answer**
Therefore, the final answer is: $\boxed{19}$. I hope it is correct.

### J.3.1 BASE ANSWER (WRONG)

Answer: To solve the problem, we are given a non-convex 11-gon $ A_1A_2\
    **dots** A_{11} $ with the following properties:

- For each $ 2 \**leq** i \**leq** 10 $, the area of triangle $ A_iA_1A_{i+1} $
    is 1.
- For each $ 2 \**leq** i \**leq** 10 $, $ \**cos**(\**angle** A_iA_1A_{i+1}) = \frac
    {12}{13} $.
- The perimeter of the polygon is 20.

- We are to find $ A_1A_2 + A_1A_{11} $, and express it in the form $ \frac{m\sqrt{n} - p}{q} $, where $ n $ is square-free and $ \gcd(m, p, q) = 1 $. Then compute $ m + n + p + q $.

---

### Step 1: Use the Area Formula

For triangle $ A_iA_1A_{i+1} $, the area is given by:
$$
\text{Area} = \frac{1}{2} \cdot A_1A_i \cdot A_1A_{i+1} \cdot \sin(\angle A_iA_1A_{i+1})
$$
We are given:
$$
\cos(\angle A_iA_1A_{i+1}) = \frac{12}{13} \Rightarrow \sin(\angle A_iA_1A_{i+1}) = \sqrt{1 - \left(\frac{12}{13}\right)^2} = \frac{5}{13}
$$
So:
$$
\frac{1}{2} \cdot x_i \cdot x_{i+1} \cdot \frac{5}{13} = 1 \Rightarrow x_i \cdot x_{i+1} = \frac{26}{5}
$$

---

### Step 2: Determine the Pattern of Side Lengths

Let $ x_i = A_1A_i $. Then from the above:
$$
x_i \cdot x_{i+1} = \frac{26}{5}
$$
This holds for $ i = 2 $ to $ 10 $, and it implies that the sequence $ x_i $ alternates between two values $ a $ and $ b $, such that:
$$
a \cdot b = \frac{26}{5}
$$

---

### Step 3: Use Law of Cosines to Find the Side Lengths Between Vertices

The side $ A_iA_{i+1} $ lies in triangle $ A_iA_1A_{i+1} $, and we can find its length using the Law of Cosines:
$$
A_iA_{i+1}^2 = x_i^2 + x_{i+1}^2 - 2x_i x_{i+1} \cos(\angle A_iA_1A_{i+1})
$$
Substituting $ x_i x_{i+1} = \frac{26}{5} $ and $ \cos(\theta) = \frac{12}{13} $, we get:
$$
A_iA_{i+1}^2 = a^2 + b^2 - 2 \cdot \frac{26}{5} \cdot \frac{12}{13} = a^2 + b^2 - \frac{48}{5}
$$

So all such sides $ A_iA_{i+1} $ have the same length $ c $, and:
$$
c^2 = a^2 + b^2 - \frac{48}{5}
$$

---

### Step 4: Use the Perimeter Constraint

The perimeter of the polygon is:

```
$$
P = A_1A_2 + A_2A_3 + \dots + A_{10}A_{11} + A_{11}A_1 = a + b + 9c
$$
Given that the perimeter is 20, we have:
$$
a + b + 9c = 20
$$

Let $ S = a + b $, then $ c = \frac{20 - S}{9} $. Substituting into the
    equation for $ c^2 $, we get:
$$
\left( \frac{20 - S}{9} \right)^2 = S^2 - \frac{48}{5}
$$

Multiply both sides by $ 81 $ to eliminate the denominator:
$$
(20 - S)^2 = 81S^2 - \frac{3888}{5}
$$
$$
400 - 40S + S^2 = 81S^2 - \frac{3888}{5}
$$

Bring all terms to one side:
$$
-80S^2 - 40S + 400 + \frac{3888}{5} = 0
$$

Multiply through by 5 to eliminate the fraction:
$$
-400S^2 - 200S + 2000 + 3888 = 0
\Rightarrow -400S^2 - 200S + 5888 = 0
$$

Divide through by -4:
$$
100S^2 + 50S - 1472 = 0
$$

Solve using the quadratic formula:
$$
S = \frac{-50 \pm \sqrt{50^2 + 4 \cdot 100 \cdot 1472}}{2 \cdot 100}
= \frac{-50 \pm \sqrt{2500 + 588800}}{200}
= \frac{-50 \pm \sqrt{591300}}{200}
$$

Simplify:
$$
\sqrt{591300} = \sqrt{100 \cdot 5913} = 10\sqrt{5913}
$$
But 5913 = $ 9 \cdot 657 = 9 \cdot 9 \cdot 73 = 81 \cdot 73 $, so:
$$
\sqrt{591300} = 10 \cdot 9 \cdot \sqrt{73} = 90\sqrt{73}
$$

Thus:
$$
S = \frac{-50 \pm 90\sqrt{73}}{200} = \frac{-5 \pm 9\sqrt{73}}{20}
$$

We take the positive root:
$$
S = \frac{9\sqrt{73} - 5}{20}
$$

---
```

### Step 5: Final Expression and Sum

We are to find $ A_1A_2 + A_1A_{11} = a + b = S = \frac{9\sqrt{73} - 5}{20} $

This matches the form $ \frac{m\sqrt{n} - p}{q} $, with:
- $ m = 9 $
- $ n = 73 $ (square-free)
- $ p = 5 $
- $ q = 20 $

Check $ \gcd(9, 5, 20) = 1 $

So, the final answer is:
$$
m + n + p + q = 9 + 73 + 5 + 20 = \boxed{107}
$$

---

### Final Answer

Therefore, the final answer is: $\boxed{107}$. I hope it is correct.

J.3.2   ANSWER OF CONTRASTIVE THINKING (CORRECT)

Answer: To solve the problem, we are given a non-convex 11-gon $ A_1A_2\ldots A_{11} $ with the following properties:

- For each $ 2 \le i \le 10 $, the area of triangle $ A_iA_1A_{i+1} $ is 1.
- For each $ 2 \le i \le 10 $, $ \cos(\angle A_iA_1A_{i+1}) = \frac{12}{13} $.
- The perimeter of the polygon is 20.
- We are to find $ A_1A_2 + A_1A_{11} $, and express it in the form $ \frac{m\sqrt{n} - p}{q} $, where $ n $ is square-free and $ \gcd(m, p, q) = 1 $, and then compute $ m + n + p + q $.

---

### Step 1: Use Triangle Area and Law of Cosines

For each triangle $ A_iA_1A_{i+1} $, the area is given by:

$$
\text{Area} = \frac{1}{2} \cdot |A_1A_i| \cdot |A_1A_{i+1}| \cdot \sin(\angle A_iA_1A_{i+1}) = 1
$$

We are given $ \cos(\angle A_iA_1A_{i+1}) = \frac{12}{13} $, so:

$$
\sin(\angle A_iA_1A_{i+1}) = \sqrt{1 - \left(\frac{12}{13}\right)^2} = \frac{5}{13}
$$

Substituting into the area formula:

$$
\frac{1}{2} \cdot |A_1A_i| \cdot |A_1A_{i+1}| \cdot \frac{5}{13} = 1 \Rightarrow |A_1A_i| \cdot |A_1A_{i+1}| = \frac{26}{5}
$$

Let $ x_i = |A_1A_i| $. Then for $ 2 \le i \le 10 $, we have:

$$
x_i \cdot x_{i+1} = \frac{26}{5}
$$

---

### Step 2: Use Law of Cosines to Find Side Lengths

For each triangle $ A_iA_1A_{i+1} $, the side $ A_iA_{i+1} $ can be found using the Law of Cosines:

$$
|A_iA_{i+1}|^2 = x_i^2 + x_{i+1}^2 - 2x_i x_{i+1} \cos(\angle A_iA_1A_{i+1})
$$

Substituting $ x_i x_{i+1} = \frac{26}{5} $ and $ \cos(\theta) = \frac{12}{13} $, we get:

$$
|A_iA_{i+1}|^2 = x_i^2 + x_{i+1}^2 - 2 \cdot \frac{26}{5} \cdot \frac{12}{13} = x_i^2 + x_{i+1}^2 - \frac{48}{5}
$$

So, the length of each side $ A_iA_{i+1} $ is:

$$
|A_iA_{i+1}| = \sqrt{x_i^2 + x_{i+1}^2 - \frac{48}{5}}
$$

---

### Step 3: Use the Perimeter Constraint

Let's define:

- $ x_2 = a $
- $ x_3 = x_5 = x_7 = x_9 = b $
- $ x_4 = x_6 = x_8 = x_{10} = a $
- $ x_{11} = \frac{26}{5a} $ (from $ x_{10} \cdot x_{11} = \frac{26}{5} $)

Then, the perimeter is:

$$
\text{Perimeter} = x_2 + 9 \cdot \sqrt{a^2 + b^2 - \frac{48}{5}} + x_{11}
$$

Substituting $ x_2 = a $, $ x_{11} = \frac{26}{5a} $, and $ ab = \frac{26}{5} \Rightarrow b = \frac{26}{5a} $, we get:

$$
\text{Perimeter} = a + 9 \cdot \sqrt{a^2 + \left(\frac{26}{5a}\right)^2 - \frac{48}{5}} + \frac{26}{5a}
$$

Let's define:

$$
t = a + \frac{26}{5a}
$$

Then, the expression inside the square root becomes:

```
$$
a^2 + \left(\frac{26}{5a}\right)^2 - \frac{48}{5} = t^2 - 20
$$
```

So the perimeter becomes:

```
$$
t + 9\sqrt{t^2 - 20} = 20
$$
```

---

### Step 4: Solve the Equation

We solve:

```
$$
t + 9\sqrt{t^2 - 20} = 20
$$
```

Isolate the square root:

```
$$
9\sqrt{t^2 - 20} = 20 - t
$$
```

Square both sides:

```
$$
81(t^2 - 20) = (20 - t)^2
\Rightarrow 81t^2 - 1620 = 400 - 40t + t^2
\Rightarrow 80t^2 + 40t - 2020 = 0
$$
```

Divide by 20:

```
$$
4t^2 + 2t - 101 = 0
$$
```

Solve using the quadratic formula:

```
$$
t = \frac{-2 \pm \sqrt{4 + 1616}}{8} = \frac{-2 \pm \sqrt{1620}}{8}
$$
```

Simplify $ \sqrt{1620} = \sqrt{324 \cdot 5} = 18\sqrt{5} $, so:

```
$$
t = \frac{-2 + 18\sqrt{5}}{8} = \frac{9\sqrt{5} - 1}{4}
$$
```

---

### Step 5: Final Answer

We are asked to find $ A_1A_2 + A_1A_{11} = x_2 + x_{11} = a + \frac{26}{5a} = t = \frac{9\sqrt{5} - 1}{4} $

This is in the form $ \frac{m\sqrt{n} - p}{q} $, with:

- $ m = 9 $
- $ n = 5 $ (square-free)
- $ p = 1 $

```
- $ q = 4 $

So:

$$
m + n + p + q = 9 + 5 + 1 + 4 = \boxed{19}
$$

---

### Final Answer

$$
\boxed{19}
$$
```

