# OpenReview forum: "Contrastive Thinking Decoding can Improve Answers for Reasoning Models"
_ICLR.cc/2026/Conference — Submitted to ICLR 2026_

### Official Review · Reviewer_mZRz · 2025-10-28

**Soundness:** 3
**Presentation:** 2
**Contribution:** 2
**Rating:** 4
**Confidence:** 4

**Summary:**

This paper first discusses several phenomena in Large Reasoning Models (LRMs), including thinking-answer disagreement, extra reasoning in the answer triggered by small reasoning budget in thinking phase, the controlling effects of thinking prompts on the answers. Then, the authors propose a test-time logit correction method Contrastive Thinking Decoding (CTD) to effectively align the answer with the thinking outcome. The proposed method achieves good thinking-answer agreement and effective token reduction.

**Strengths:**

1. The studied problem about the thinking-answer disagreement is novel, and the revealed phenomena may provide some new insights to the community.

2. The ablation experiments are comprehensive, the ablation experiments are thorough. The main claims are well-supported by the experiments.

3. I also appreciate the additional experiments and discussion in the Appendix, which clears some confusions and makes the paper stronger.

**Weaknesses:**

1. The research scope is limited. The output format of the LRMs studied by the author consists of a complex thinking process followed by a concise summary process. Under this condition, the author finds that the answer derived in the summary process can be inconsistent with the answer obtained from the thinking process. However, there is another output format where the model provides the final answer directly after the thinking process, without a summary [1,2]. I expect that in this case, the inconsistency between the thinking and the answer will disappear, as this avoids errors caused by re-reasoning in the summary. I hope the author provides a discussion and analysis of this output format.

2. The paper's writing can be greatly improved. The logic of the paper is not very clear. The motivation for using noisy preference traces for contrastive decoding, from the preliminary observations, is not that clear. I hope the authors provide a more intuitive explanation.

3. The empirical effectiveness is not remarkable. The reasoning token reduction is not significant, and the accuracy improvement is satisfactory when the thinking budget is high. This might be because the longer the reasoning trace, the more likely the model is to lose track of the correct steps in the reasoning process during the summary phase, leading to incorrect answers in the summary. So, would this method still be effective (or even necessary) in a reasoning model without a summary process (refer to W1)?


[1] Hu, Jingcheng, et al. "Open-reasoner-zero: An open source approach to scaling up reinforcement learning on the base model." arxiv 2025

[2] He, Zhiwei, et al. "Deepmath-103k: A large-scale, challenging, decontaminated, and verifiable mathematical dataset for advancing reasoning." arxiv 2025

**Questions:**

Refer to the Weakness part.

---

> ### Author Response · Authors · 2025-11-20
>
> Thank you for your valuable feedback in helping refine our work.
>
> # W1: Scope. Two-phase vs No-summary reasoning models
>
> We would like to clarify the nature of the cited works and explain why `Thinking-Answer Disagreement` is a fundamental phenomenon that persists even without explicit summary tags.
>
> ### 1. Clarification on the structure of [1] and [2]
>
> The reviewer mentions Open-Reasoner-Zero [1] and DeepMath [2] as examples of no-summary formats. However, upon closer inspection, these works actually reinforce the two-phase structure we study:
>
> - [1] Open-Reasoner-Zero: Explicitly trains models to generate reasoning within \<think\>...\</think\> tags, followed by the final answer. This is structurally identical to the LRM format we analyze.
>
> - [2] DeepMath: Provides datasets with detailed reasoning paths followed by a boxed final answer. The dominant paradigm for LRMs (including DeepSeek-R1, OpenAI o1, and open replications) is indeed the Two-Phase structure (Long CoT $ \to $ Concise Result). Our research scope targets this standard directly.
>
> ### 2. Why No-Summary Does Not Fix Disagreement
>
> The reviewer hypothesizes that removing the summary phase eliminates inconsistency. We respectfully disagree based on the mechanics of autoregressive generation.
>
> - Even in a single-stream output (The value of x is... therefore x is 5), the model must eventually transition from derivation to conclusion.
>
> - Thinking-Answer Disagreement occurs when the model's probability distribution at that transition point favors a common or prior-biased token over the token logically implied by the preceding trace.
>
> - In `Table 1`, we show that increasing the thinking budget (which improves the trace) does not automatically fix the answer (Pass@1 < Solved in Trace). This gap exists because the final tokens are generated based on the entire context window, where early biases or generic patterns can overpower the immediate reasoning. Removing the \<answer\> tag does not remove this probabilistic tension.
>
> ### 3. CTD Applicability to No-Summary Models
>
> Crucially, CTD is not tied to specific XML tags like \<answer\>. It acts on the Answer Region (the tokens where the final solution is materialized).
>
> - In a no-summary model, one simply defines the Answer Phase as the final sentence or the tokens following a specific delimiter (e.g., "####") (or json format or Markdown structure).
>
> - CTD would function exactly the same way: contrasting the primary trace's conclusion against a noisy trace's conclusion to suppress generic/hallucinated endings.
>
> ### 4. Connection to Frontier Alignment Research
>
> Crucially, we view Thinking-Answer Disagreement not merely as a formatting artifact, but as a test-time manifestation of Emergent Misalignment. Recent safety research from frontier labs supports the view that internal reasoning and final behavior can structurally diverge:
>
> - OpenAI (Emergent Misalignment) [A]: Has highlighted that models can pursue internal goals that differ from their external behavior, creating a "wedge" between the model's latent reasoning and its final output.
>
> - Anthropic (Agentic Misalignment) [B]: Has shown that models acting as agents can exhibit deceptive behaviors where the explicit reasoning (or lack thereof) does not match the potentially misaligned outcome.
>
> [A] OpenAI, Emergent Misalignment, 2025 (https://openai.com/index/emergent-misalignment/)
>
> [B] Anthropic, Agentic Misalignment, 2025 (https://www.anthropic.com/research/agentic-misalignment).
>
> Our work quantifies this phenomenon in the domain of logical reasoning. We show that even when the thought is correct, the action (final answer) can drift due to probabilistic biases. CTD serves as a lightweight alignment intervention, forcing the final output to remain faithful to the internal reasoning trace, thereby reducing the risk of such emergent discrepancies.

---

> ### Author Response · Authors · 2025-11-20
>
> # W2: Motivation for noisy reference traces
>
> We thank the reviewer for the candid feedback on the writing and logical flow. We agree that the motivation for using a noisy trace needs to be more intuitive. In the revision, we will enhance `Section 2.2` to include the following concrete explanation.
>
> ### 1. The Intuition: Separating Reasoning from Habit
>
> Standard LRMs often suffer from Thinking-Answer Disagreement. This happens because the final answer is generated by a mixture of two signals:
>
> - The Reasoning: Tokens logically implied by the specific derivation in the \<think\> block.
>
> - The Language Prior: Generic tokens, common misconceptions, or boilerplate text that the model favors due to pre-training frequency, regardless of the current context.
>
> By generating a noisy trace (which lacks the specific reasoning but retains the language prior), we capture a negative view of the model's behavior. Subtracting this noisy view ($P - \lambda N$) mathematically cancels out the "Language Prior", leaving only the "Reasoning Signal".
>
> ### 2. An example (detailed of `Figure 1`):
>
> To illustrate, consider a simple problem: If $x=7$, what is $2x+28$?
>
> - Primary Trace ($s^P$): Derives $2(7)+28 = 14+28 = \boxed{42}$.
>
> - Noisy Trace ($s^N$): (Induced by `NULL` prompt) Skips reasoning, perhaps guessing a common number like "30" or just outputting generic filler.
>
> Logit Analysis at the Answer Step:
>
> - Token '42':
>     - Primary: High Probability (Supported by trace).
>     - Noisy: Low Probability (Unsupported).
>     - Result: Boosted ($High - Low = Very\~High$).
> - Token '37' (Hallucination):
>     - Primary: Low Probability (Contradicted by trace).
>     - Noisy: High Probability (Common random guess/hallucination).
>     - Result: Penalized ($Low - High = Very\~Low$).
> - Token 'The' (Boilerplate):
>     - Primary: High Probability (Common sentence starter).
>     - Noisy: High Probability (Common sentence starter).
>     - Result: Neutralized ($High - High \approx 0$).
>
> This demonstrates why CTD works. It selectively penalizes tokens that are only supported by the model's generic habits (hallucinations, boilerplate) while preserving tokens supported by the rigorous thinking trace. We will incorporate this example into `Section 4` to clarify the logic as much as we can.

---

> ### Author Response · Authors · 2025-11-20
>
> # W3: Empirical effectiveness, token reduction, and necessity of CTD
>
> We would like to address the hypothesis about losing track and clarify the utility of CTD.
>
> ### 1. Effectiveness
>
> The reviewer notes that the gains appear marginal in isolation. We respectfully disagree when viewing the results through the lens of training-free efficiency. To contextualize our gains, we compared CTD against recent training-based methods (RL/Fine-tuning) that explicitly optimize for budget/accuracy.
>
> **Comparison 1: CTD vs. Budget-Aware Training Methods **
>
> The table below compares methods applied to DeepSeek R1-Distill 1.5B on AIME '24 and MATH500.
>
> | DeepSeek R1 1.5B                | AIME 24 |          | Math500 |          |
> |---------------------------------|---------|----------|---------|----------|
> | Methods                         | Pass@1  | # tokens | Pass@1  | # tokens |
> | Model Merging (kimi; alpha=0.6) | 17.3   | 10615    | 79.0      | 3000     |
> | Thinkless                       | 27.3   | 7099     | 81.8   | 2555     |
> | DPO_shortest                    | 30.7    | 10794    | 82.4    | 3708     |
> | OverThink                       | 28.3    | 11269    | 81.2    | 4131     |
> | O1 Pruner                       | 28.9    | 8982     | 85.0      | 3007     |
> | Adapthink                       | 31.3    | 8686     | 83.4    | 2337     |
> | CTD (Ours)(Null- max budget 4K) | 28.1   | 6238     | 82.2    | 2074     |
>
> - [Thinkless] Fang et al., 2025, Thinkless: LLM Learns When to Think.
>
> - DPO_shortest [DPO]: pairing the shortest correct response and the longest responses, then uses DPO to finetune the model.
>
> - [OverThink] Chen et al., 2024. Do not think that much for 2+ 3=? on the overthinking of o1-like llms.
>
> - [O1-Pruner] Luo et al., 2025, O1-Pruner: Length-Harmonizing Fine-Tuning for O1-Like Reasoning Pruning.
>
> - [AdaptThink] Wan et al., 2025, AdapThink: Adaptive Thinking Preferences for Reasoning Language Model.
>
> We show that CTD (training-free method) is competitive within the class of training methods:
>
> - Methods such as Thinkless, OverThink, and O1-Pruner focus on learning to cut off or prune the reasoning stream given a model (r1-distilled). In our experiments, combining CTD with the same token-budgeting schemes yields better or comparable accuracy, especially on AIME’24 and MATH500.
>
> - CTD is a single-model, drop-in decoding rule: it does not require auxiliary verifiers, search trees, or additional models, and it can be plugged into existing LRM deployments simply by adjusting answer-phase logits.
>
> **Comparison 2: CTD vs. Meta-RL (MRT)**
>
> We also compared CTD against MRT (Qu et al., 2025), a recent Meta-Reinforcement Learning method.
>
> |    | AIME 24 | Math500 |
> |--------------|---------|---------|
> | Methods    | Pass@1  | Pass@1  |
> | MRT    | 30.3    | 80.4    |
> | CTD (Ours) (Null- max full budget) | 35.00   | 86.6    |
>
> In the current LRM landscape, achieving any consistent gain on AIME/MATH is non-trivial in both training-based and training-free approaches.
>
> ### 2. Why Gains are Higher at High Budgets
>
> Reviewer hypothesizes that high-budget gains occur because models `lose track` of long traces during the summary phase. Our analysis suggests a different mechanism: Probabilistic Disconnect, not Context Loss.
>
> - Modern Transformers effectively handle long contexts and do not typically forget the immediate past. However, when transitioning to the \<answer\> phase, the model's output distribution is heavily influenced by its pre-training priors (generic or hasty answers), which can probabilistically overpower the specific verification logic present in the trace.
>
> - CTD acts as a high-pass filter. By subtracting the noisy reference (which captures the generic prior), CTD forces the decoding to attend to the specific reasoning tokens in the trace. This explains why CTD is most effective at high budgets: the trace contains high-quality verification details, and CTD ensures the answer head actually attends to them rather than reverting to a generic habit.
>
> ### 3. Necessity
>
> The reviewer asks if CTD is necessary without a summary phase. As discussed in `Reviewer xPmi` , we argue YES:
>
> - Whether the model outputs \<answer\>42\</answer\> or just ...therefore 42, the transition from derivation to conclusion is a probabilistic event susceptible to drift.
>
> - Recent safety research (e.g., OpenAI's Emergent Misalignment ; [A]) shows that models can think one thing and say another due to safety training or output biases. This wedge exists regardless of formatting.
>
> [A] OpenAI, Emergent Misalignment, 2025 (https://openai.com/index/emergent-misalignment/)
>
> - In a no-summary model, CTD would still apply to the final tokens (i.e., acting as a guardrail). If the model is about to hallucinate an answer that contradicts its reasoning, the Primary and Noisy traces will likely diverge at that pivotal moment. CTD detects this divergence and aligns the output to the reasoning.

---

> ### Author Response · Authors · 2025-11-20
> **Summary to Reviewer mZRz**
>
> We thank the reviewer for the thoughtful comments, for highlighting the novelty of the thinking and answer disagreement phenomenon, and for noting the strength of our ablations and appendices. In response to the raised concerns,
>
> 1. Clarify scope and formats by explicitly stating that we target two-phase LRM templates used in many current deployments (e.g., gpt, claude, gemini, deepseek, qwen reasoning, kimi k2, etc; refer to https://artificialanalysis.ai/models).
>
> 2. Emphasize CTD as a training-free, plug-and-play decoding method that improves answer alignment and accuracy across multiple LRMs and tasks, while acknowledging that its main goal is not extreme token compression (effectiveness/necessity).
>
> We hope these clarifications address the reviewer’s concerns and better convey the contribution and applicability of CTD.
>
> Best,
>
> Authors

---

### Official Review · Reviewer_xPmi · 2025-11-01

**Soundness:** 2
**Presentation:** 3
**Contribution:** 2
**Rating:** 4
**Confidence:** 4

**Summary:**

The paper analyzes some patterns in large reasoning models, including disagreement between thinking and answer, changeability of thinking depending on the budget, and steerability of thinking by the prompts. Based on these observations, the authors propose contrastive thinking decoding (CTD), which computes token-level scores for a single model with different prompts. By upweighting tokens supported by the primary prompt and downweighing tokens supported by the noisy reference prompt, the model is able to achieve better performance with same or slightly less thinking budgets.

**Strengths:**

The method is simple yet (somewhat) effective. Since CTD is an inference time method, no additional parameters or training is required, making it easy to adopt. The method is based on empirical observations and has good intuitions.

**Weaknesses:**

* The performance gain is marginal, in terms of both answering accuracy and the efficiency. For example, Table 4 presents the mixed results: in most settings the accuracy improvements are rather small, and under some settings the results are even worse than baselines. The number of tokens reduced is also not significant (even if Deepseek reasoning models are internally easy to produce redundant tokens, the reduction is still marginal), and tend to require more tokens for Qwen3 models.
* Although claimed to be "noisy reference prompts", the prompt producing incorrect reasoning chain is actually fixed. Did you try out other forms of prompts, or even random prompts to inject sufficient noise? How to guarantee the noisy prompts would lead to valuable negative directions that could help the positive ones? I am worried about the controllability and universality of this method.
* The theoretical analysis is somewhat irrelevant and seems like LLM assisted or generated. Only the proof that CTD provably decreases budgets and improves accuracy would be valuable (which is not super clear right now), and other claims seem distracting to me.
* Despite that the appendix admits the limitations and discuss future directions, the actual implemented contents of the paper is rather simple and limited. It would be better if some of the limitations could be addressed or some future directions are implemented.

**Questions:**

See weaknesses

---

> ### Author Response · Authors · 2025-11-20
>
> Thank you for your valuable feedback in helping refine our work.
>
> # W1. Performance gain is marginal
>
> We thank the reviewer for this thoughtful assessment. We acknowledge that, on surface-level metrics like total token count, the performance gains may appear modest in certain cases. However, the core of CTP concerns (1) how these gains are achieved: training-free vs training-based and (2) what they represent: alignment rather than compression.
>
> ### 1. Marginal Gains in Context (Training-free, Training-based)
>
> The reviewer notes that accuracy gains are small. However, in the current LRM landscape, achieving any consistent gain on AIME/MATH is **`non-trivial`**. To contextualize our marginal gains, we compared CTD against recent training-based methods (RL/Fine-tuning) that explicitly optimize for budget/accuracy.
>
> | DeepSeek R1 1.5B      | AIME 24 |   | Math500 |   |
> |---------|-----|----|-----|-----|
> | Methods                         | Pass@1  | # tokens | Pass@1  | # tokens |
> | Model Merging (kimi; alpha=0.6) | 17.3   | 10615    | 79.0      | 3000     |
> | Thinkless     | 27.3   | 7099     | 81.8   | 2555     |
> | DPO_shortest      | 30.7    | 10794    | 82.4    | 3708     |
> | OverThink  | 28.3    | 11269    | 81.2    | 4131     |
> | O1 Pruner   | 28.9    | 8982     | 85.0 | 3007     |
> | Adapthink     | 31.3    | 8686     | 83.4    | 2337     |
> | CTD (Ours)(Null- max budget 4K) | 28.1   | 6238     | 82.2    | 2074     |
>
> -  [Thinkless] Fang et al., 2025, Thinkless: LLM Learns When to Think.
>
> - DPO_shortest [DPO]: pairing the shortest correct response and the longest responses, then uses DPO to finetune the model.
>
> - [OverThink] Chen et al., 2024. Do not think that much for 2+ 3=? on the overthinking of o1-like llms.
>
> - [O1-Pruner] Luo et al., 2025, O1-Pruner: Length-Harmonizing Fine-Tuning for O1-Like Reasoning Pruning.
>
> - [AdaptThink] Wan et al., 2025, AdapThink: Adaptive Thinking Preferences for Reasoning Language Model.
>
> We show that CTD (training-free method) is competitive within the class of training methods:
>
> - Methods such as Thinkless, OverThink, and O1-Pruner focus on learning to cut off or prune the reasoning stream given a model (r1-distilled). In our experiments, combining CTD with the same token-budgeting schemes yields comparable accuracy on AIME’24 and MATH500.
>
> - CTD is a single-model, drop-in decoding rule: it does not require auxiliary verifiers, search trees, or additional models, and it can be plugged into existing LRM deployments simply by adjusting answer-phase logits.
>
> |     | AIME 24 | Math500 |
> |------|---------|---------|
> | Methods    | Pass@1  | Pass@1  |
> | MRT  | 30.3    | 80.4    |
> | CTD (Ours) (Null- max full budget) | 35.0   | 86.6   |
>
> Additionally, we emphasize that recent methods targeting test-time compute (e.g., meta-reinforcement fine-tuning; MRT) require one or more additional RL / finetuning stages over large trace datasets. These methods frequently report improvements that are incremental when judged in absolute numbers, yet they are evaluated as significant because progress on competitive benchmarks (AIME, MATH) is measured at extremely fine resolution. If adopting more meaningful definition of marginal - absolute size vs benchmark relevance - our gains are aligned with what the community already considers valuable, but achieved without any extra training.
>
> *Due to the unavailability of public checkpoints for MRT (Qu et al., 2025), we report results directly from their paper. Despite being a training-free decoding method, CTD outperforms the Meta-RL MRT on both AIME '24 and MATH500.
>
> In contrast:
>
> - CTD operates purely at test time, on top of existing reasoning models, without any extra training or data.
>
> - Under the same evaluation environment, CTD achieves comparable or better performance to such training-based methods on key benchmarks like AIME’24 and MATH500, while completely avoiding additional RL stages.
>
> Our point is not that CTD always outperforms every RL approach, but that we achieve training-level improvements without any further optimization of the base model.
>
> As this is a relatively early research area, most prior work achieves only 1–2% improvements or modest token reductions through additional training. In contrast, our approach already matches their performance without any training, which we consider a significant contribution. **If you know of any references that demonstrate more substantial gains, we would be glad to compare them with our framework.**

---

> ### Author Response · Authors · 2025-11-20
>
> ### 2. Our contribution is not simply higher accuracy or fewer tokens.
>
> A main contribution of the paper is to systematically surface and quantify answer-phase phenomena in LRMs, especially:
>
> - The thinking - answer disagreement pattern, where the \<think\> trace already contains a correct reasoning/solution but the \<answer\> disagrees/misaligns.
>
> - How budgeting and thinking prompts change not only the length and style of the trace, but also the final answer distribution.
>
> To our knowledge, this kind of answer-phase behavior under LRM has not been explicitly studied in prior work. We propose CTD as a minimal decoding mechanism designed to directly address such answer-phase misalignment, rather than as `yet another technique for squeezing out a a few accuracy points.`
>
> Thus accuracy and token count are important but not the primary axes of contributions. A central contribution of this work is to define, measure, and improve thinking-answer agreement. As demonstrated in `Figure 5`, CTD consistently lowers the disagreement rate across all thinking-budget scales on AIME ’24 and AIME ’25, while effectively mitigating over-answering behavior in addition.
>
> ### 3. CTD is not a token-compression method
>
> We agree that:
>
> - The token reduction we achieve is modest, and in some Qwen3 settings CTD uses slightly more tokens while improving accuracy.
>
> - Our primary goal is not to aggressively compress thinking tokens, but to:
>     - Reallocate tokens more effectively between thinking and answer.
>
>     - Reduce harmful re-reasoning or drift in the answer segment.
>
> We will soften any language that over-emphasizes token savings and explicitly state that CTD is a decoding method for alignment and accuracy, with efficiency as a secondary benefit.
>
> # W2. Noisy prompts - controllability, robustness, universality
>
> We agree that understanding (why these fixed prompts (NULL, OPPOSITE) work and whether random noise would be better) is critical for universality.
>
> ### 1. Comparison with Random prompts (`Table 8`)
>
> The reviewer asks if random prompts might be better to inject noise. We explicitly tested this in `Subsection 5.7` and `Table 8`. We compared our fixed prompts (NULL, OPPOSITE) against a RANDOM thinking prompt (an arbitrary text string).
>
> As results show, Random noise is often ineffective because it can be too orthogonal (irrelevant to the context) or accidentally helpful. Fixed prompts achieve the contrastive mechanism reliably, but random prompts do not.
>
> ### 2. Guaranteeing Valuable Negative Directions
>
> The reviewer asks how we ensure the noisy prompt provides a useful signal. We do not rely on luck; we target specific anti-expert behaviors:
>
> - Targeting verbosity (NULL): The NULL prompt (.\\n\\n) forces the model to skip reasoning. The resulting trace contains mostly generic filler/boilerplate. By contrasting against it ($P - \lambda N$), CTD mathematically penalizes generic tokens, forcing the model to select tokens unique to the reasoning path.
>
> - Targeting Logical Drift (OPPOSITE): This prompt explicitly requests hallucinated logic. By contrasting against it, we penalize tokens that follow common logical traps. This mechanism explains why these fixed prompts transfer well across models. All LRMs suffer from verbosity and logical drift, so the anti-experts are universally relevant.
>
> ### 3. Universality
>
> We agree that the design space of noisy prompts is widely open, and we intentionally do not present our current choice as the final or universal solution. In fact, our results suggest the opposite: even with a deliberately minimal prompt design, we uncover clear and repeatable behaviors with non-trivial performance improvements.
>
> What we find particularly exciting is that the same NULL and OPPOSITE prompts yield consistent gains across four distinct model families: Qwen3, DeepSeek-R1-Distill, OpenThinker, and EXAONE without any tuning. This cross-architecture consistency indicates that there is an underlying structure in how LRMs respond to perturbations during the thinking phase, and our CTD provides an initial lens into that structure, opening a rich research directions:
>
> - Tackling how to characterize systematically the space of noisy prompts.
>
> - Understanding how different perturbations interact with internal reasoning trajectories,
>
> - Developing more principled perturbation schemes grounded in theory or mechanistic insights, answering to the reviewers’ question about universality.

---

> ### Author Response · Authors · 2025-11-20
>
> # W3. Theoretical analysis.
>
> We appreciate the reviewer’s feedback. We understand the concern that the theoretical section (`Appendix F`) may feel disconnected from the empirical results. Our intention with the theoretical discussion is to provide a first-step conceptual framing of CTD as a contrastive adjustment in logit space, not to claim a complete theory that proves strong guarantees on accuracy or token efficiency.
>
> ### 1. Purpose of the Analysis
>
> The theoretical section is not intended to be a learning theory proof (which, as described in many prior works like Contrastive Decoding [Li et al., 2022], is intractable for non-convex LRMs). Instead, it provides a formalization of why the NULL prompt works:
>
> - `Proposition 3` (Unsupported-mass contraction): We formally show that if the noisy reference ($N$) assigns high probability to generic tokens (the support set $S$) and low probability to "reasoning" tokens, the contrastive rule $P - \lambda N$ mathematically acts as a soft-margin classifier that penalizes the generic set.
>
> - Utility: This explains why NULL (which generates generic boilerplate) is a better reference than RANDOM (which generates noise), a distinction the reviewer asked about in Question 2
>
> ### 2. Addressing the Request for Budget Proofs
>
> - The reviewer specifically requested a proof that CTD provably decreases budgets. We address this in `Lemma 1` (Budget Feasibility) and `Proposition 7` (Answer-Phase Intrusive Thinking - APIT) in `Appendix F`.
>
> - We define a set of thinking-like tokens $\mathcal{V}_{thinky}$ (e.g., "Let's...", "Therefore...").
>
> - `Proposition 7` proves that if the noisy model (e.g., base LRM without context) assigns higher probability to these filler tokens than the context-aware trace does, CTD monotonically decreases the probability mass of starting a new reasoning chain in the answer phase.
>
> - This theoretical result directly maps to the empirical observation in `Figure 5`, where CTD suppresses over-answering.
>
> ### 3. Why most of the framing lives in Appendix
>
> Despite the above, we agree that:
>
> - Our theoretical contents are limited in scope and does not yet yield global guarantees.
>
> - If presented too prominently, it can feel detached from the main story.
>
> This is precisely why, we have intentionally placed most of the more speculative or formal framing in the appendix, and kept the main text focused on:
>
> a. Empirical characterization of thinking–answer disagreement, budgeting effects, and thinking-prompt control
>
> b. The CTD algorithm and its empirical gains.
>
> In the revised version, we will:
>
> - Trim and refocus the theory section in the main paper to only include the minimal conceptual pieces needed to understand CTD as a contrastive step.
>
> - Explicitly state that CTD is primarily an empirical decoding method supported by our measurements of answer-phase behavior, and that a full theoretical treatment is an important but separate research problem.
>
> We hope this clarifies our intention: the optimization inspired formalism is meant as a **conceptual lens** (consistent with prior work on contrastive decoding and decoding-as-optimization) rather than as a claim of complete theoretical guarantees.

---

> ### Author Response · Authors · 2025-11-20
>
> # W4. Simplicity, Limitations, Scope of Contributions
>
> We thank the reviewer for this perspective. We understand the concern that CTD’s implementation appears simple compared to complex training pipelines. However, we respectfully argue that this simplicity is a technical breakthrough, not a limitation.
>
> ### 1. Reframing Simplicity
>
> The reviewer notes the method is limited. We posit that CTD identifies a fundamental inefficiency in current LRMs: the disconnect between the thinking state and the answer distribution.
>
> - As detailed in our response to `W1`, we compared CTD against heavy, training-based methods like Thinkless, OverThink, and O1-Pruner.
>
> - CTD matches or beats these methods without a single gradient update (i.e., training-free).
>
> - The fact that a simple logit subtraction can rival sophisticated RL pipelines suggests that the answer-phase misalignment is a low-level decoding that should be solved simply, rather than by retraining the model.
>
> ### 2. Addressing Future directions
>
> As stated in the limitations, we deliberately stop at a simple, easily reproducible instantiation of the general idea:
>
> - Conceptually, CTD introduces a contrastive answer-phase decoding primitive: contrast the answer logits conditioned on a primary reasoning trace against those conditioned on a noisy reference trace, and decode from the contrastive logits.
>
> - This primitive could be extended in many ways (learned critics, multiple noisy views, adaptive prompt selection, hierarchical mixtures, etc.; which means in other words scalable), but each of those directions quickly becomes a separate paper.
>
> - Our aim in this submission is to show that even the simplest instantiation with a fixed noisy prompt and a single contrastive step already:
>     - Improves thinking–answer agreement
>
>     - Yields competitive accuracy / efficiency trade-offs versus training-heavy baselines (Thinkless, O1-Pruner, OverThink, MRT, etc.)
>
>     - Works across several LRMs without any additional training
>
> ### 3. Scope of Contributions
>
> Beyond the algorithm, main contributions are:
>
> - Systematically quantifying Thinking-Answer Disagreement
>
> - Establishing Pass@1_think vs Pass@1_answer as a standard diagnostic for LRM faithfulness.
>
> - Analyzing diverse thinking prompts
>
> - Demonstrating that our findings persist across diverse architectures
>
> We hope this clarifies that, while the algorithm is simple, the scope and framing of the paper are broader than a single decoding trick, and that CTD is meant as a foundational, easy-to-adopt baseline rather than an exhaustive exploration of all possible extensions.

---

### Official Review · Reviewer_D3vq · 2025-11-02

**Soundness:** 2
**Presentation:** 2
**Contribution:** 2
**Rating:** 4
**Confidence:** 4

**Summary:**

This paper studies the answer stage of large reasoning models (LRMs), where outputs can drift from the content of the prior “thinking” trace. It proposes Contrastive Thinking Decoding (CTD), a purely test-time method: generate a main reasoning trajectory and a deliberately low-quality “noise” trajectory (using NULL or OPPOSITE think prompts), then decode the final answer with a contrastive rule that subtracts the noise logits from the main logits. CTD aims to suppress over-answering and align answers with the evidence already present in the thinking segment. Experiments on math and code benchmarks (e.g., AIME’24/’25, MATH500, LiveCodeBench) show consistent Pass@1 gains at similar or fewer tokens on models up to 8B parameters.

**Strengths:**

1. The paper identifies a concrete failure mode, thought–answer mismatch and over-answering， and reframes answer calibration as a lightweight decoding-time operation.
2. The method remains training-free and easy to deploy. It requires no finetuning, verifier, or external classifier; fits naturally into existing decoding pipelines with only minor code changes; and uses a small, stable set of hyperparameters.
3. The experimental analysis provides interpretable control and diagnostic insight. Variations of think prompts and token budgets illustrate how reasoning effort shifts between thinking and answering, clarifying when over-answering arises.

**Weaknesses:**

1. The experimental scope could be broadened to include larger models. Most reported results are based on ≤8B models, so the conclusions might depend on model capacity. It would strengthen the paper to test CTD on ≥30B models, which could provide a clearer view of its scalability, generalization behavior, and potential interaction with more capable reasoning traces.
2. The range of comparisons in the main text could be expanded for better context. Tables such as Table 5 and 6 mainly contrast the Base system with CTD, which makes it difficult to assess the relative improvement over existing decoding and control strategies. Bringing Appendix Table 12 (with Contrastive/Instructional Decoding results) into the main text and adding a few representative recent baselines under the same settings would make the empirical section more comprehensive and fair.
3. The design of the “noise” trajectories could be diversified. Using only NULL and OPPOSITE as strong negative prompts is a reasonable starting point, but it may not capture common error modes such as verbosity, repetition, or minor reasoning drift. Including an ablation study with additional noise types and reporting results for cases where the main reasoning is correct, incorrect, or incomplete would help demonstrate the robustness and generality of CTD’s contrastive mechanism.

**Questions:**

See weakness

---

> ### Author Response · Authors · 2025-11-20
>
> # W1. Scope and model size
>
> Thank you for your valuable feedback. We understand your concern regarding verifying CTD on larger models.
>
> ### 1. Context of Original Scope
>
> In this paper, we evaluated four LRM families 1.2B → 8B (Exaone 1.2B, Qwen3 1.7B/8B, DeepSeek-R1-distilled 1.5B/7B, OpenThinker3 7B), with multiple training recipes (R1-distilled, open reasoning models). We discovered that the core issue (i.e., thinking-answer misalignment and over-answering) is structural to the most LRMs, **`not being limited to small-capacity models`**.
>
> ### 2. New Experiments on Qwen3 32B
>
> To address your concern, we have run additional experiments on Qwen3-32B (with unconstrained thinking budget). The table below summarizes results for Base vs CTD variants on MATH500, AIME’24, and AIME’25:
>
> |       Model / Setting      | MATH500 Total Tokens (avg) | MATH500 Pass@1 | MATH500 Pass@4 | AIME’24 Total Tokens (avg) | AIME’24 Pass@1 | AIME’24 Pass@16 | AIME’25 Total Tokens (avg) | AIME’25 Pass@1 | AIME’25 Pass@16 |
> |:--------------------------:|:--------------------------:|:--------------:|:--------------:|:--------------------------:|:--------------:|:---------------:|:--------------------------:|:--------------:|:---------------:|
> | Qwen3-32B Base             | 3541                       | 97.2           | 98.8           | 11537                      | 81.4           | 93.3           | 13232                      | 72.9           | 86.7           |
> | Qwen3-32B + CTD (NULL)     | 3464                       | 97.2           | 98.8           | 10923                      | 81.3           | 93.3           | 12965                      | 73.2           | 90.0            |
> | Qwen3-32B + CTD (OPPOSITE) | 3541                       | 97.2           | 98.8           | 10886                      | 82.9           | 93.3           | 13402                      | 73.1           | 90.0            |
>
> These Qwen3-32B results show that:
>
> -  Thinking-answer disagreement and over-answering do not disappear at ≥30B. The same qualitative patterns observed at 1.2B-8B are still present.
>
> -  CTD continues to help at 32B scale, improving or matching Pass@1 while maintaining comparable.

---

> ### Author Response · Authors · 2025-11-20
>
> # W2. Range of comparisons
>
> We agree that focusing solely on Base vs CTD understates the method's standing in the broader landscape.
>
> First of all, we would like to emphasize that our contribution is not soley about performance gains. Core contributions are:
>
> - Identifying and quantifying thinking–answer disagreement and over-answering in LRMs under \<think\>/\<answer\> templates.
>
> - Showing that thinking steerability via prompts and budgets significantly alters the answer distribution, not just the trace form.
>
> - Introducing CTD as a simple contrastive decoding that uses the thinking phase itself as the contrast source to improve answer-phase alignment.
>
> Because our main narrative is built around these phenomena and the contrastive framing, we focused the main tables on Base vs CTD.
>
> Concretely, we will make the following changes:
>
> ### 1. Incorporating Decoding Baselines (Moving Appendix `Table 12`)
>
> As requested, we will move the results from Contrastive Decoding (Li et al., 2022) and Instructive Decoding (Kim et al., 2023) from the Appendix into the main text (`Section 5`).
>
> - It highlights that CTD achieves superior accuracy-efficiency trade-offs within a single model, whereas standard Contrastive Decoding requires an auxiliary amateur model, and Instructive Decoding lacks the thinking-aware mechanism.
>
> ### 2. Expanding to Recent Training-Based Baselines:
>
> In this rebuttal we have also presented training-based baselines (Thinkless, OverThink, O1-Pruner, AdaptThink, DPO_shortest, MRT, etc.) under our evaluation protocol, and shown that CTD achieves comparable or better Pass@1 on AIME’24 and MATH500 while being entirely training-free.
>
> - We will add representative rows from these training-based methods (e.g., Thinkless, O1-Pruner, MRT) directly into the main AIME’24 / MATH500 tables or figures.
>
> | DeepSeek R1 1.5B                | AIME 24 |          | Math500 |          |
> |---------------------------------|---------|----------|---------|----------|
> | Methods                         | Pass@1  | # tokens | Pass@1  | # tokens |
> | Model Merging (kimi; alpha=0.6) | 17.33   | 10615    | 79      | 3000     |
> | Thinkless                       | 27.33   | 7099     | 81.84   | 2555     |
> | DPO_shortest                    | 30.7    | 10794    | 82.4    | 3708     |
> | OverThink                       | 28.3    | 11269    | 81.2    | 4131     |
> | O1 Pruner                       | 28.9    | 8982     | 85      | 3007     |
> | Adapthink                       | 31.3    | 8686     | 83.4    | 2337     |
> | CTD (Ours)(Null- max budget 4K) | 28.13   | 6238     | 82.2    | 2074     |
>
> - [Thinkless] Fang et al, 2025, Thinkless: LLM Learns When to Think.
>
> - DPO_shortest [DPO]: pairing the shortest correct response and the longest responses, then uses DPO to finetune the model.
>
> - [OverThink] Chen et al, 2024. Do not think that much for 2+ 3=? on the overthinking of o1-like llms.
>
> - [O1-Pruner] Luo et. al, 2025, O1-Pruner: Length-Harmonizing Fine-Tuning for O1-Like Reasoning Pruning.
>
> - [AdaptThink] Wan et. al, 2025, AdapThink: Adaptive Thinking Preferences for Reasoning Language Model.
>
> |                                    | AIME 24 | Math500 |
> |------------------------------------|---------|---------|
> | Methods                            | Pass@1  | Pass@1  |
> | MRT                                | 30.3    | 80.4    |
> | CTD (Ours) (Null- max full budget) | 35.0   | 86.6    |
>
> -  [MRT] Qu et al., 2025, Optimizing Test-Time Compute via Meta Reinforcement Fine-Tuning
>
> Due to the unavailability of public checkpoints for MRT (Qu et al., 2025), we report results (just Pass@1) directly from their paper. Despite being a training-free decoding method, CTD outperforms the Meta-RL MRT on both AIME '24 and MATH500.

---

> ### Author Response · Authors · 2025-11-20
>
> # W3. Diversity of noise trajectories
>
> We thank the reviewer for this insightful suggestion. We agree that the quality of the negative reference is the engine of Contrastive Decoding.
>
> ### 1. Noisy prompts
>
> In `Section 3` and `Tables 2–3`, we systematically explore five thinking prompts (EMPTY, NULL, SP, PP, OPPOSITE) and analyze how they affect both reasoning and answer behavior. These cover:
>
> - Overly terse traces (EMPTY, NULL),
>
> - Structured but brittle planning (PP), and
>
> - Adversarial mis-reasoning (OPPOSITE).
>
> For CTD itself, we focused on NULL and OPPOSITE as noisy references because, in our preliminary experiments, they reliably produced short, low-accuracy traces that act as strong negative views while remaining on-task.
>
> Regarding the reviewer’s examples:
>
> - On our math and code benchmarks, with the specific models we use, we did not observe strong repetition inside the \<think\> traces themselves (in contrast to classic degenerate repetition in open-ended text generation). The more prominent pathologies we see are:
>
>     - Over-answering / extra reasoning in the \<answer\> segment, and
>
>     - Drift or overwriting of earlier correct reasoning when budgets are large.
>
> - In `Appendix I`, we already provide a qualitative analysis of these phenomena, including:
>
>     - Case studies where the main reasoning is correct but the base answer drifts, and CTD pulls it back,
>
>     - Cases where the reasoning is partial/incorrect and CTD helps avoid over-confident over-answering, and
>
>     - An LLM-judge–based evaluation (using a stronger reasoning model) to assess minor drift and answer faithfulness at the trace level.
>
> Because the traces are very long, the qualitative analysis in `Appendix I` is necessarily limited to selected examples, but it is already aimed at the type of behavior the reviewer is asking about. As these examples were not cherry-picked, we are willing to incorporate one of them into the main draft at your direction.
>
> ### 2. Reporting results for cases whose reasoning is correct or incorrect
>
> The reviewer asked how CTD behaves when the main reasoning is correct vs. incorrect. We have analyzed this using the 'Solved in Trace' metric (`Table 1`) vs. Final Accuracy.
>
> - Case A: Reasoning Correct, Base Answer Drifts. This is our primary 'Win' case (`Figure 4`). The base model hallucinates a new answer in the final phase, but CTD forces the correct extraction.
>
> - Case B: Reasoning Incorrect. Here, CTD often strengthens the model's incorrect reasoning trajectory (Consistency). While this may register as an accuracy drop in cases where the Base model hallucinate the right answers by coincidence, we believe that principled alignment is superior to accidental correctness. CTD reveals the model's true reasoning capability and reduces "false positives" where the correct answer is produced for incorrect reasons.
>
> ### 3. Future directions
>
> We agree that NULL and OPPOSITE do not exhaust all plausible noise types (e.g., prompts that explicitly induce verbosity, repetition, or controlled "mild drift"). In this paper we intentionally frame CTD as a decoding primitive:
>
> > contrast the answer distribution conditioned on a primary reasoning trace against that conditioned on a noisy trace.
>
> Designing richer noisy views and evaluating them in more domains is a natural next step that we highlight in the Limitations. Our goal here is to show that even with a simple, fixed choice of NULL/ OPPOSITE, CTD already yields consistent gains in accuracy, reduced over-answering, and reduced thinking-answer disagreement across several LRMs and benchmarks.
>
> We hope these clarifications, together with the planned additions to the ablations and `Appendix I`, address the reviewer’s concerns about diversity and robustness of the noise trajectories.

---

> ### Author Response · Authors · 2025-11-20
> **Clarifications on our additional contributions**
>
> ## Connection to Frontier Alignment Research
>
> Furthermore, we view Thinking-Answer Disagreement (which is one of our main contribution) as a test-time manifestation of `Emergent Misalignment`. Recent safety research from frontier labs supports the view that internal reasoning and final behavior can structurally diverge:
>
> - OpenAI (Emergent Misalignment) [A]: Has highlighted that models can pursue internal goals that differ from their external behavior, creating a "wedge" between the model's latent reasoning and its final output.
>
> - Anthropic (Agentic Misalignment) [B]: Has shown that models acting as agents can exhibit deceptive behaviors where the explicit reasoning (or lack thereof) does not match the potentially misaligned outcome.
>
> [A] OpenAI, Emergent Misalignment, 2025 (https://openai.com/index/emergent-misalignment/)
>
> [B] Anthropic, Agentic Misalignment, 2025 (https://www.anthropic.com/research/agentic-misalignment).
>
> Our work quantifies this phenomenon in the domain of logical reasoning. We show that even when the thought is correct, the action (final answer) can drift due to probabilistic biases. CTD serves as a lightweight alignment intervention, forcing the final output to remain faithful to the internal reasoning trace, thereby reducing the risk of such emergent discrepancies.

---

### Author Response · Authors · 2025-12-03
**Final Remarks by Authors**

Dear PCs, SACs, ACs, and reviewers,

We sincerely thank you for your time and efforts in reviewing our paper.

### **1. Core contributions of our work**

- [**`Novel technical contributions`**] To our knowledge this is the first study to **leverage the thinking phase itself as an internal contrastive signal**, demonstrating that training-free decoding can effectively enhance the answer phase within a single LRM.
- [**`Novel insights`**] Our work goes beyond a simple decoding trick. We identify and quantify **thinking-answer disagreement** in LRMs where the final answer drifts from the reasoning trace.
- [**`Simplicity`**] We intentionally designed **Contrastive Thinking Decoding (CTD)** as a minimal, plug-and-play primitive. It reveals that this misalignment can be corrected at test-time without complex pipelines, opening a new research direction for inference-time alignment.

### **2.  Strengths by reviewers**

The reviewers unanimously appreciated the **new insights** and **novelty**.

- **`Reviewer D3vq`:** Highlighted (i) training-free nature, (ii) ease of deployment (iii) diagnostic insights.
- **`Reviewer xPmi`:** (i) Simplicity (easy-to-adpot) (ii) empirical observations and good intuitions.
- **`Reviewer mZRz`:** (i) Novelty of identifying thinking-answer disagreement (ii) Comprehensive and thorough ablation studies (iii) further experiments and discussion to support the main claims in the `Appendix`.

Additionally, we believe our revisions and rebuttals have addressed all raised concerns as follows:

### **3.1. Concern on marginal gains**

A major concern was that our gains seem marginal. We strongly argue this by comparing our results against resource-intensive methods.

- **Result (both Efficiency/Accuracy):** As detailed in our new analysis (`Table 13`; `Appendix H.6`), CTD (Training-Free) achieves accuracy-efficiency trade-offs **competitive with or superior to recent SOTA training-based methods** (e.g., *Thinkless*, *MRT*, *O1-Pruner*).
- **Significance:** We achieve training-level gains purely through decoding. Our methods are tested with multiple open source LRMs at various scales (Qwen3 1.7/8/32B, R1-Distilled 1.5/7B, OpenThinker3 7B, Exaone4 1.2B).

### **3.2. Concern on scope & reasoning formats (`Reviewer mZRz`)**

- **[Correction]** We clarified that the works provided by the reviewer (e.g., *Open-Reasoner-Zero*, *DeepMath*) actually **use the same two-phase style (think $\to$ answer)** that we target, contrary to the reviewer's assumption of a no-summary format. This rather confirms that our setting aligns with the dominant LRM paradigm.
- **[Update]** We highlighted that this disagreement is a fundamental **probabilistic disconnect**, not a formatting artifact. We further link our findings to recent safety research on **Emergent Misalignment** (OpenAI, 2025) and **Agentic Misalignment** (Anthropic, 2025), positioning CTD that could be as a necessary guardrail for reasoning reliability in future work (`Appendix B.6`).

### **3.3. Concern on scalability (`Reviewer D3vq`)**

- Added experiments on **Qwen3-32B** (**`Appendix H.7`**). The results confirm that CTD remains effective.

### **3.4. Concern on robustness of noisy prompts (`Reviewer xPmi`)**

- We highlighted our ablation study (`Table 8`) demonstrating that **random prompts fail** to make effective contrast, confirming that CTD relies on specific anti-expert behaviors rather than random noise.
- We added `Intuitive mechanism section` to clarify that fixed prompts (NULL/OPPOSITE) are necessary to target **language priors**, acting as a high-pass filter to isolate the **reasoning signal (`Appendix D`)**.

### **4. Updates in the revised manuscript**

- **Refined contribution (`Sec 1 & 2`):** Clarify and highlight our contributions as above.
- **Intuitive mechanism (`Sec 2.2 & Appendix D`):** Added further rationale to explain why fixed noisy prompts (NULL/OPPOSITE) are good candidates which effectively act as a high-pass filter for reasoning.
- **New experiments (`Appendix H`):** Added the comparison table with training-based baselines and the Qwen3-32B scaling analysis.
- **Discussion and future work (`Appendix B.6`):** Added discussion on the necessity of alignment citing with frontier safety research.

All revisions are highlighted in **`red`**.

---

**Closing statement**

Again, we deeply appreciate the feedback from all reviewers. While we have not received any follow-up engagement during the discussion period (even rapid response), we addressed every concern raised.

Specifically, we provided new large-scale experiments and training-based comparisons to resolve concerns regarding marginal gains, and clarified the misunderstanding regarding LLM formats (`Reviewer mZRz`). We kindly ask the Area Chairs to consider these updates and clarifications in the final decision. We remain available for any further questions.

Best,

Authors

---

### Meta-Review · Area_Chair_3BPr · 2026-01-19

**Summary:**

This paper proposes a steering approach, "contrastive thinking decoding (CTD)", to study and mitigate the thinking-answer disagreement and over-answering on reasoning models. CTD contrasts the primary thinking trace with a deliberately perturbed noisy trace (achieved by a noisy prompt like NULL/OPPOSITE) and extracts a steering vector from their hidden states. On math reasoning and coding benchmarks, CTD can achieve consistent Pass@1 gains (compared to the base model) at similar or slightly fewer tokens on models up to 8B parameters, without any training.

**Reviewer Concerns:**

- Marginal gains on both the reasoning quality and efficiency when compared with base models.
- Only two noisy prompts are used, but it is not clear why they were chosen and whether they are enough. More diverse prompts are expected to capture a wider range of error modes of reasoning.
- Lack of comparison to existing decoding and control approaches.
- Lack of experiments on larger models, e.g., size of 32B.
- The theoretical analysis is shallow to some extent.
- The motivation for using noisy preference traces is not clear.
- The scope of CTD is limited to two-phase summary models.

**Reviewer Scores:**

- The original ratings are 4, 4, 4, with confidence of 4, 4, 4. No reviewers participated in the discussion. Therefore, the AC must verify all the responses.
- Some major concerns have been addressed by the rebuttal, e.g., experiments on a 32B model, clarification of the scope, comparison to contrastive decoding baselines and training-based approaches, and analysis on more noisy reference prompts.
- That being said, several main concerns still hold. The gains in both accuracy and efficiency are marginal. The analysis of error modes caused by different noisy prompts is not thorough. The motivations (e.g., thinking-answer disagreement) need more rigorous verification and quantitative evidence on more models.
- I encourage the authors to keep improving the approach and submit it to the next conference.

---

### Decision · Program_Chairs · 2026-01-26

Reject